# A fungal sesquiterpene biosynthesis gene cluster critical for mutualist-pathogen transition in *Colletotrichum tofieldiae*

Kei Hiruma [1,2] ✉, Seishiro Aoki [3], Junya Takino[4], Takeshi Higa[1],
Yuniar Devi Utami [1], Akito Shiina[1], Masanori Okamoto [5], Masami Nakamura[1],
Nanami Kawamura [2], Yoshihiro Ohmori[6], Ryohei Sugita [7], Keitaro Tanoi [6],
Toyozo Sato[8], Hideaki Oikawa [9], Atsushi Minami [4], Wataru Iwasaki [3] &
Yusuke Saijo [2]

Plant-associated fungi show diverse lifestyles from pathogenic to mutualistic to the host; however, the principles and mechanisms through which they shift the lifestyles require elucidation. The root fungus *Colletotrichum tofieldiae* (Ct) promotes *Arabidopsis thaliana* growth under phosphate limiting conditions. Here we describe a Ct strain, designated Ct3, that severely inhibits plant growth. Ct3 pathogenesis occurs through activation of host abscisic acid pathways via a fungal secondary metabolism gene cluster related to the bio-synthesis of sesquiterpene metabolites, including botrydial. Cluster activation during root infection suppresses host nutrient uptake-related genes and changes mineral contents, suggesting a role in manipulating host nutrition state. Conversely, disruption or environmental suppression of the cluster renders Ct3 beneficial for plant growth, in a manner dependent on host phosphate starvation response regulators. Our findings indicate that a fungal metabolism cluster provides a means by which infectious fungi modulate lifestyles along the parasitic–mutualistic continuum in fluctuating environments.

Plants associate intimately with diverse microbes, including pathogens, nonpathogenic commensals, and beneficial (mutualistic) microbes that promote plant growth. Plant–microbe interactions are typically context-dependent, as these microbes dynamically change their infection modes according to the environment and host conditions[1,2]. One microbe can switch between pathogenic and beneficial infection modes even in the same host, depending on the host and environmental conditions, without changing the microbial genomic sequences[3,4]. These findings imply that different lifestyles of plant-associated microbes are continuous within the same plant species and even coexist within the plant individual. Moreover, these microbes possess the capacity to refine infection strategies according to the given host environments[5,6]. However, the mechanisms by which infectious microbes transit between their contrasting lifestyles remain elusive.

[1]Department of Life Sciences, Graduate School of Arts and Sciences, The University of Tokyo, 3-8-1, Komaba, Meguro-ku, Tokyo 153-8902, Japan. [2]Department of Science and Technology, Nara Institute of Science and Technology, Nara 630-0192, Japan. [3]Department of Integrated Biosciences, Graduate School of Frontier Sciences, The University of Tokyo, Chiba 277-0882, Japan. [4]Department of Chemistry, Faculty of Science, Hokkaido University, Kita 10, Nishi 8, Kita-ku, Sapporo 060-0810, Japan. [5]Center for Bioscience Research and Education, Utsunomiya University, 350 Mine-cho, Utsunomiya, Tochigi 321-8505, Japan. [6]Graduate School of Agricultural and Life Sciences, The University of Tokyo, 1-1-1, Yayoi, Bunkyo-ku, Tokyo 113–8657, Japan. [7]Radioisotope Research Center, Nagoya University, Furo-cho, Chikusa-ku, Nagoya 464-8602, Japan. [8]Genetic Resources Center, National Agriculture and Food Research Organization, Ibaraki 305-8602, Japan. [9]Innovation Center of Marine Biotechnology and Pharmaceuticals, School of Biotechnology and Health Sciences, Wuyi University, Jiang-men, Guangdong 529020, China. ✉e-mail: hiruma@g.ecc.u-tokyo.ac.jp

Plants have evolved an elaborate system called phosphate starvation response (PSR) to cope with low inorganic phosphate (Pi) conditions[7]. Under low Pi, *A. thaliana* plants induce extensive transcriptome reprogramming, mainly through the R2R3-MYB family transcription factors *PHOSPHATE RESPONSE1* (*AtPHR1*) and related *PHR1-LIKE1* (*AtPHL1*)[8,9]. Plant adaptation to Pi deficit involves the activation of genes that promote phosphate absorption, allocation, and usage. *A. thaliana* plants also accommodate beneficial root-associated endophytic fungus *Colletotrichum tofieldiae* (Ct) that helps nutrient acquisition. Their hyphae acquire and transfer phosphorus to the host, providing an extension of the plant root system[3]. *AtPHR1* and *AtPHL1* positively regulate phosphate transporter genes during Ct colonization and are required for Ct-mediated plant growth promotion under low Pi[3]. *AtPHR1/AtPHL1* restricts fungal overgrowth and potential virulence during beneficial interactions with Ct[3], whereas negatively regulating plant immunity against bacteria[10,11]. However, our knowledge on the mechanisms by which *AtPHR1/AtPHL1* play varied roles in different plant-microbe associations or host PSR influences microbial lifestyles is limited.

Microbes have evolved an enormous repository of secondary metabolites, including plant hormone mimics for auxin and gibberellic and abscisic acid (ABA). Whereas more than half of the genes are organized in operons in bacteria, functionally related genes are typically distributed across the genome in eukaryotic fungi[12]. However, fungal secondary metabolite biosynthetic genes, as well as their regulatory genes, are often clustered in genetic loci[13,14]. The necrotrophic fungal pathogen, *Botrytis cinerea*, harbors a cluster of biosynthetic genes for ABA and produces ABA in culture[15-17]. There is a good correlation between the fungal ABA production and the host ABA signaling activation during *B. cinerea* infection, which facilitates the suppression of immune-related genes in *A. thaliana*[18,19]. However, the expression patterns of fungal ABA biosynthesis genes during plant infection remain elusive. The possible significance also remains to be explored for these ABA and/or other secondary metabolite biosynthesis genes in the activation of host ABA pathways and fungal virulence. Furthermore, these secondary metabolism genes are often not expressed even during host interactions in conventional laboratory settings[20,21]. Consistently, disruptions of individual genes in secondary metabolism clusters typically do not alter fungal phenotypes *in-planta*[22]. These limitations have hampered the precise determination of the roles of secondary metabolism clusters in plant-infecting fungi.

In the present study, we describe a pathogenic strain, Ct3, causing severe plant growth inhibition, in contrast to beneficial strains prevailing in Ct. Comparative genomics and functional analyses between these Ct strains indicate that, following root colonization, Ct3 displays transcriptional activation of biosynthesis gene clusters, putatively for ABA and the associated sesquiterpene metabolite botrydial (designated ABA-BOT), thereby promoting root infection and pathogenesis. Conversely, Ct3 colonization gives the host benefits when this fungal cluster is genetically disrupted or transcriptionally suppressed at mildly elevated temperatures in a manner dependent on *AtPHR1/AtPHL1*. These findings highlight the key role of fungal secondary metabolites in the infection-mode transition of plant-associated fungi.

## Results

### A Ct strain severely inhibits plant growth in a nutrient-dependent manner

A Ct strain, Ct61, isolated from a wild *A. thaliana* population in Spain, promotes plant growth under low Pi conditions by transferring phosphorus to the host[3]. In addition to Ct61, five different Ct strains have been isolated from various plant species and geographical locations (Supplementary Table 1)[23,24]. Molecular phylogenetic analysis using six molecular markers conventionally utilized for the identification of *Colletotrichum* species[21,23-26] is consistent with the view that the tested

six Ct strains belong to the same species (Supplementary Fig. 1). Notably, however, these five strains exhibited different growth morphology during in vitro growth (Fig. 1a and Supplementary Fig. 2a).

This prompted us to investigate the possible intraspecies variations of Ct in plant infection effects and strategies. We compared six Ct strains in their inoculation effects on *A. thaliana* Col-0 under gnotobiotic low Pi conditions (50 μM KH$_2$PO$_4$). Five of the tested Ct strains, including Ct61, significantly promoted plant growth, indicated by primary root length and shoot fresh weight (Fig. 1b, c and Supplementary Fig. 2b, c). The results suggest that plant growth-promoting (PGP) function under low Pi, at least discernible in *A. thaliana*, is extensively shared by Ct strains from various host and geographical niches. Notably, in contrast to these PGP strains, Ct3 severely inhibited shoot growth under low Pi (Fig. 1b, c and Supplementary Fig. 2b, c). Ct3 inhibited plant growth in additional 15 *A. thaliana* accessions (Supplementary Fig. 2d) and also in *Brassica rapa* var. *perviridis* (Supplementary Fig. 2f, Left), whose growth was promoted by beneficial Ct4 (Supplementary Fig. 2f, Right), on unsterilized low nutrient soil. These results suggest that Ct3 pathogenic lifestyle is common in a broad diversity of Brassicaceae species, at least under the tested conditions.

Since Ct61 PGP function is specific to low Pi[3], we tested whether phosphate availability influences Ct3 and Ct4 infection phenotypes. Under normal Pi conditions (625 μM), Ct3 caused plant growth inhibition and leaf chlorosis (Fig. 1d and Supplementary Fig. 2c, g), whereas Ct4 did not cause plant growth inhibition or leaf chlorosis (Fig. 1c, d). Under low Pi, Ct3 compromised plant growth but did not cause leaf chlorosis or reduce leaf chlorophyll contents (Fig. 1c, d; Supplementary Fig. 2c, g). In contrast to Ct3, Ct4 promoted plant growth (Fig. 1d). Since sucrose increased and accelerated Ct3 growth but not Ct4 growth in culture (Supplementary Fig. 2h), we hypothesize that Ct3 pathogenesis is dependent on host photosynthate supply and is alleviated when its supply is likely reduced under phosphate deficiency.

To trace fungal colonization dynamics, we generated transgenic fungal strains expressing cytoplasmic GFP (Ct3-GFP or Ct4-GFP) under constitutive *GPDA* regulatory DNA sequences[3]. Conidia of the transgenic Ct strains were then inoculated onto *A. thaliana* roots, expressing an aquaporin PIP2A fused with mCherry, a plasma membrane/ER marker in living host cells[3]. Up to 5 days post-inoculation (dpi), both Ct3 and Ct4 hyphae both reached the cortex cell layer in the root (Fig. 1e, f, Supplementary Movie 1, Supplementary Movie 2). The intracellular hyphae of Ct3 and Ct4 were weakly labeled with the mCherry-expressing plant plasma membrane, indicating that these Ct strains colonize inside the roots of *A. thaliana*, as previously described for Ct61[3]. The results indicate that Ct3 and Ct4 are either pathogenic or beneficial fungi colonizing *A. thaliana*, respectively, under the tested conditions.

To grasp the genomic basis for the intraspecies variations between pathogenic Ct3 and beneficial Ct4, we determined their whole-genome sequences using a long-read generating PacBio sequencing platform. PacBio long reads provided high-quality genome assemblies for all strains, ranging from 53 to 55 Mb, with similar gene numbers for candidate effector proteins, carbohydrate-active enzymes, and secondary metabolism clusters (Supplementary Table 2). Whole-genome alignment between Ct3 and Ct4 showed a high degree (Median: >98.6%) of nucleotide identity between the two genomes, despite their contrast infection strategies. In contrast, a comparison between the beneficial strains Ct61 and Ct4 showed numerous genomic rearrangements and a lesser degree of nucleotide identity (88.9%), implying a genome-wide divergence between Ct61 and Ct4, despite their similarities in host benefits (Fig. 2a). These results suggest that nucleotide identity scores largely reflect the geographical distances of their origins. However, a molecular phylogenetic analysis using the conserved 1509 single-copy genes among the tested 71 fungal species suggests that beneficial Ct strains have evolved

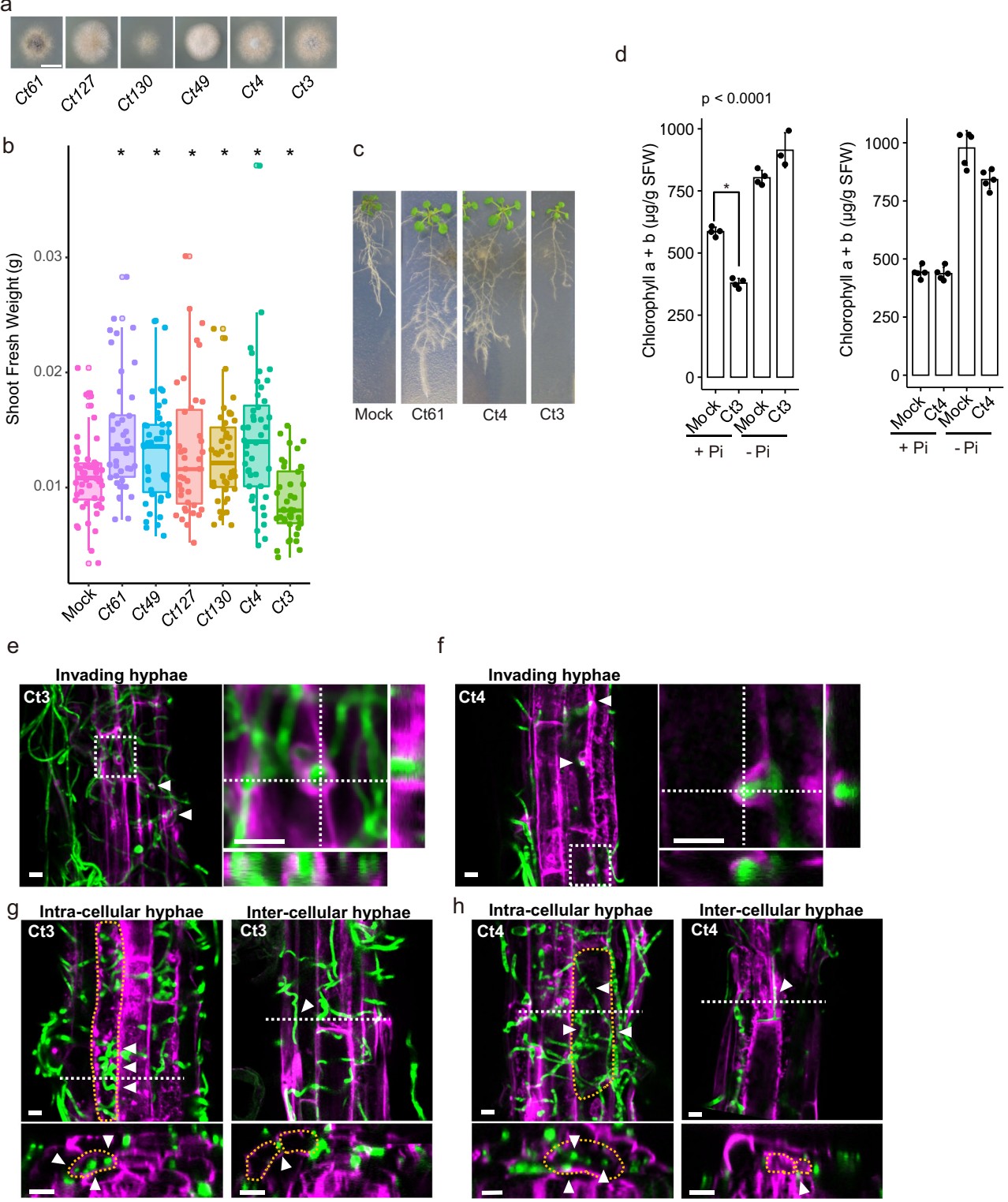

from their pathogenic relatives, such as *C. incanum* (Fig. 2b; Supplementary Data 1). It seems likely that beneficial lifestyles have evolved in pathogenic ancestors of Ct species.

## Pathogenic Ct3 activates host PSRs and ABA signaling pathways during early root infection

We next investigated plant responses following root inoculation with beneficial and pathogenic Ct strains via RNA-seq analysis at two-time points, 10 dpi and 24 dpi (Fig. 3a). At 10 dpi, Ct3 caused plant growth inhibition under both normal and low Pi conditions, whereas PGP

effects of beneficial Ct strains were not yet apparent (Supplementary Fig. 3a). At 24 dpi, the degree of plant growth inhibition by Ct3 was lowered under Pi deficiency compared with Pi sufficiency (Supplementary Fig. 3b), whereas PGP by beneficial Ct61 and Ct4 became discernible only under Pi deficiency (Fig. 1b and Supplementary Fig. 3b). In contrast to these Ct strains, the KHC strain (previously classified as *C. tofieldiae*), closely related to *C. higginsianum* (Fig. 2b and Supplementary Fig. 1), did not alter plant growth under low Pi condition (Fig. 3a and Supplementary Fig. 3b), displaying a non-pathogenic lifestyle.

**Fig. 1 | Root infection of a *Colletotrichum tofieldiae* strain (Ct3) severely inhibits plant growth in *A. thaliana*. a** Growth of Ct strains identified from various geographical locations in Mathur's nutrient media after 3 days of incubation. Bar = 1 cm. **b** Determination of *A. thaliana* shoots fresh weight following fungal inoculation under low Pi, 24 days after seed germination in the presence of the indicated fungal strains. Boxplot represents combined results from three independent experiments. Each dot represents individual plant samples (Mock: *n* = 51, Ct61: *n* = 42, Ct49: *n* = 43, Ct127: *n* = 39, Ct130: *n* = 44, Ct4: *n* = 44, Ct3: *n* = 41 biologically independent samples). The median values are described within each boxplot. Asterisks indicate significantly different means between mock and fungal-inoculated plants (*p* < 0.01, two-tailed *t*-test). **c** *A. thaliana* plants grown in normal or low Pi conditions with or without fungal inoculation (24 dpi). **d** Shoot chlorophyll a and b contents with or without Ct3 or Ct4 inoculation under normal and low Pi (*n* = 3 for Ct3_Low Pi or *n* = 4 for others biologically independent samples). Asterisks indicate significantly

different means between mock and Ct3-inoculated plants (±SD, *p* < 0.0001, two-tailed *t*-test). **e–h** Confocal microscopic images of Ct3 (**e**, **g**) or Ct4 (**f**, **h**) expressing cytoplasmic GFP (green) and root cells of *A. thaliana* expressing PIP2A-mCherry (magenta). These experiments were repeated three times with similar results. Maximum projection of z-stack images shows Ct3 (**e**) or Ct4 (**f**) hypha penetrating a root epidermal cell surrounded by PIP2A-mCherry-labeled host membranes (arrowheads). Representatives of hyphal penetrations are shown as enlarged projected images of different optical sections and orthogonal views made from areas indicated by white dotted line. Maximum projection of z-stack images shows Ct3 (**g**) or Ct4 (**h**) hypha elongating intra- or inter-cellular spaces. Intra- or inter-cellular hyphae were indicated by white arrowheads and orthogonal views were made from areas indicated by white dotted line. Cells allowing elongation of intra-cellular hypha or contacting with inter-cellular hypha were surrounded by a yellow dotted line. Bar = 10 μm.

We first focused on the divergence in plant responses between these Ct strains at 10 dpi, when Ct3 already caused plant growth inhibition (Supplementary Fig. 3a). Multidimensional scaling analysis indicated a clear separation of plant transcriptome in response to Ct3 compared to beneficial Ct and nonpathogenic KHC strains, suggesting distinct plant responses to Ct3 at this stage (Fig. 3b). The results revealed 758 *A. thaliana* genes that were differentially upregulated in Ct3-infected roots compared with beneficial Ct strains, under normal or low Pi conditions (log$_2$FC> 1, FDR < 0.05, Supplementary Data 2). *K*-means clustering classified these differentially expressed genes (DEGs) into three different clusters (Fig. 3c and Supplementary Data 3). In Clusters 1 and 2, Gene Ontology (GO) categories related to ABA signaling were overrepresented (Fig. 3c and Supplementary Data 3), suggesting that Ct3 specifically induces host ABA responses. ABA promotes plant adaptation to water deficit[27] and influences immunity or susceptibility-related pathways[28]. qRT-PCR analyses validated the increased expression of ABA-responsive plant genes, *AtRD29A*[29] and *AtMAPKKK18*[30], during pathogenic Ct3 infection (Fig. 3d, e). ABA accumulation also increased in roots following Ct3, but not Ct4, colonization (Fig. 3f). The results indicate that Ct3, but not Ct4, strongly induces ABA responses in roots. We then tested whether and how host ABA contributes to Ct3-mediated plant growth inhibition, with plant mutants disrupted in ABA signaling, *abi1-1C*; ABA biosynthesis, *aba1* and *aba2-12*; and ABA perception, *pyr1-1 pyl1 pyl2 pyl4 pyl5*. Ct3-mediated plant growth inhibition was alleviated in these ABA-defective mutants under low Pi (Fig. 3g). In contrast, plant growth promotion via beneficial Ct4 under low Pi was not affected in *abi1-1C* (Supplementary Fig. 3c). These analyses suggest that pathogenic but not beneficial lifestyles of Ct3 are dependent on the core ABA biosynthetic and response pathways of the host.

During beneficial interactions under low Pi, Ct61 induced a subset of PSR-related genes in plants, including phosphate transporter *AtPHT* genes, at 24 dpi[3]. Since PSR-related transcriptional reprogramming largely relies on *AtPHR1* and *AtPHL1*[9], we next assessed the possible impact of beneficial Ct strains on *AtPHR1/AtPHL1*-regulated PSR-related genes under low Pi at 24 dpi when PGP by beneficial Ct61 and Ct4 became discernible. Cross-referencing our data with previously described 193 *AtPHR1/AtPHL1* regulons[9] indicated increased expression for approximately half of them [88 of 193 genes (Ct61), or 99 of 193 genes (Ct4), FDR < 0.05] during beneficial Ct interactions compared with the mock controls, specifically under low Pi (Supplementary Fig. 3d). In contrast, PSR gene activation was not increased during pathogenic or nonpathogenic interaction with Ct3 or KHC, respectively, at 24 dpi (Supplementary Fig. 3d). At 10 dpi, however, Ct3 strongly upregulated a subset of PSR-related genes under low Pi compared with the mock controls (Supplementary Fig. 3e, 146 of 193 genes, FDR < 0.05), indicated by the induction of Clusters 2 and 3 genes related to PSRs, in which GOs overrepresented "Cellular response to starvation (Cluster 2)" and "cellular response to phosphate starvation (Cluster 3)" (Fig. 3c and Supplementary Data 3).

Upregulation of PSR genes was associated with an increase in shoot P concentrations following Ct3 inoculation under low Pi (Supplementary Fig. 3f), coincident with alleviated pathogenesis. Notably, Ct3 inoculation upregulated a subset of *AtPHR1/AtPHL1* regulons, which were otherwise not induced, under Pi sufficient conditions (Fig. 3c (Cluster 3) and Supplementary Fig. 3e, 69 of 193 genes, FDR < 0.05). The results suggested that Ct3 infection results in *AtPHR1/AtPHL1* regulon activation, at least during an early infection phase, despite the eventual negative effects on plant growth.

We then examined how *AtPHR1/AtPHL1*-dependent PSR influences pathogenic Ct3 lifestyles by testing for Ct3 inoculation phenotypes in *phr1 phl1* plants. Compared to WT, *phr1 phl1* displayed severe growth inhibition following Ct3 inoculation under low Pi (Supplementary Fig. 3g). The results indicate a critical role for *AtPHR1/AtPHL1* in the alleviation of plant growth inhibition under low Pi. Together with the transcriptome analyses above (Supplementary Fig. 3), we infer from the results that activation of *AtPHR1/AtPHL1* regulons serves to restrict Ct3 pathogenesis.

**Putative ABA and BOT biosynthesis gene clusters are distributed across plant-associated fungi, possibly along with fungal virulence**

To assess the basis for transcriptional activation of host ABA responses, we examined fungal transcriptome profiles during pathogenic Ct3 infection (Supplementary Data 4). As Ct3-mediated ABA response activation and plant growth inhibition were pronounced at 10 dpi (Fig. 3c and Supplementary Fig. 2g), we assembled Ct3 genes that were differentially upregulated at 10 dpi compared to 24 dpi. This resulted in 304 fungal genes upregulated in an early phase under normal or low Pi conditions (Fig. 4a, log$_2$FC > 1, FDR < 0.05). The number of these genes was greater under normal Pi than low Pi (Fig. 4a), consistent with enhanced Ct3 impacts on plant growth under normal Pi (Fig. 1).

Interestingly, 92 among 304 Ct3 genes were only expressed at 10 dpi under both Pi conditions, but their expression was below detectable levels at 24 dpi, coincident with fungal pathogenesis that was pronounced at 10 dpi (Supplementary Data 5). We noticed that putative ABA biosynthesis genes (*Ct3ABA1*, *Ct3ABA2*, and *Ct3ABA3*) were included in the 92 pathogenesis-associated genes. They show high sequence similarity to ABA biosynthesis genes (*BcABA1*-*BcABA3*) in *B. cinerea* (Supplementary Data 4). *BcABA1* and *BcABA2* encode cytochrome P450, and *BcABA3* encodes sesquiterpene synthase catalyzing the initial step from farnesyl diphosphate (FPP), in a fungal ABA biosynthesis pathway that is entirely different from that of plants[16,17,31]. Interestingly, in Ct3, these putative ABA biosynthesis genes are clustered together with those highly related to the biosynthesis of botrydial (BOT), an ABA-related sesquiterpene metabolite in *B. cinerea* (Fig. 4b, Supplementary Table 3)[32,33]. Five BOT biosynthesis genes (named *BOT1*-*BOT5*) were co-activated with the three putative ABA biosynthesis genes during root colonization in Ct3 but not in other beneficial Ct strains (Fig. 4c, Supplementary Table 4), despite the high

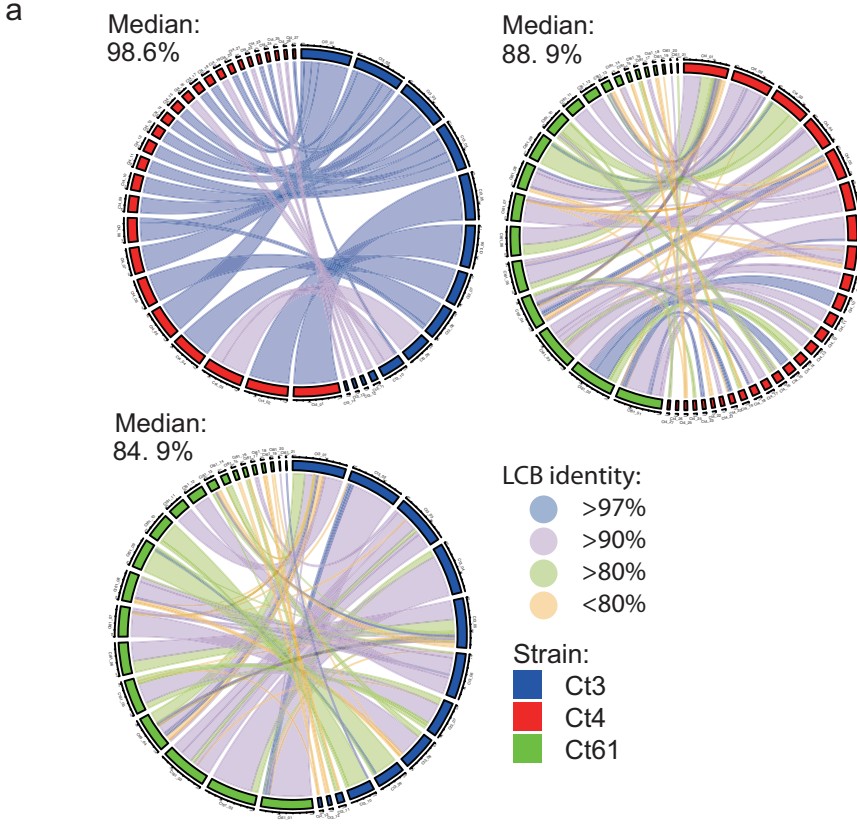

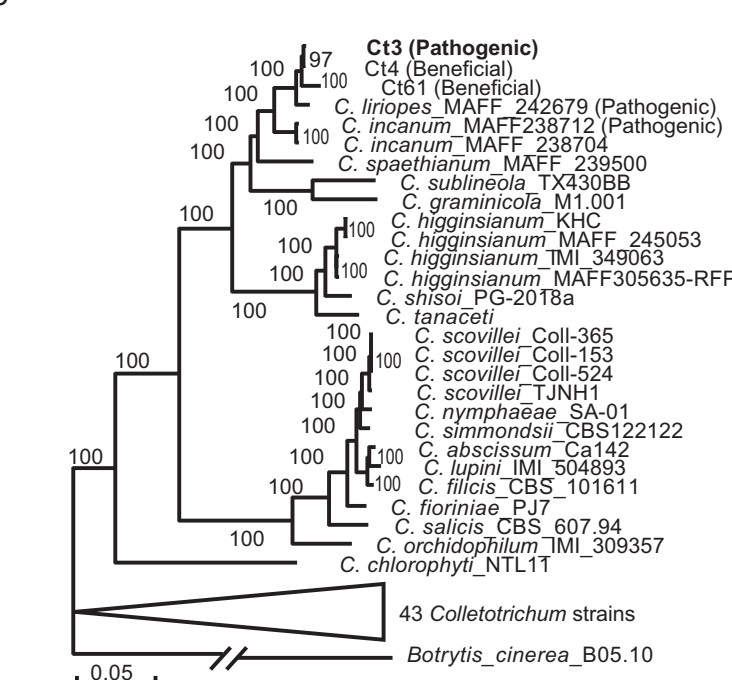

**Fig. 2 | Genomic analyses of beneficial and pathogenic Ct strains. a** Syntenies of Ct3, Ct4, and Ct61 genomes. The length of each contig was described as 0.8 Mb/scale. The percentage in the figure represents the median % of the nucleotide similarity against the whole locally collinear block (LCB). Pairwise genome comparisons point to the highest similarity between Ct3 and Ct4 and the lowest between Ct61 and Ct3. **b** Maximum-likelihood phylogenomic tree generated from a concatenated alignment of 1509 single-copy orthologous protein sequences from 70 *Colletotrichum* strains using IQ-TREE. Ultrafast bootstrap values (1000 replicates) are shown on the branches. *B. cinerea* B5.10 was used as an outgroup. The lifestyles of some *Colletotrichum* fungi in *A. thaliana* roots under low Pi were investigated and are described in the tree (beneficial or pathogenic).

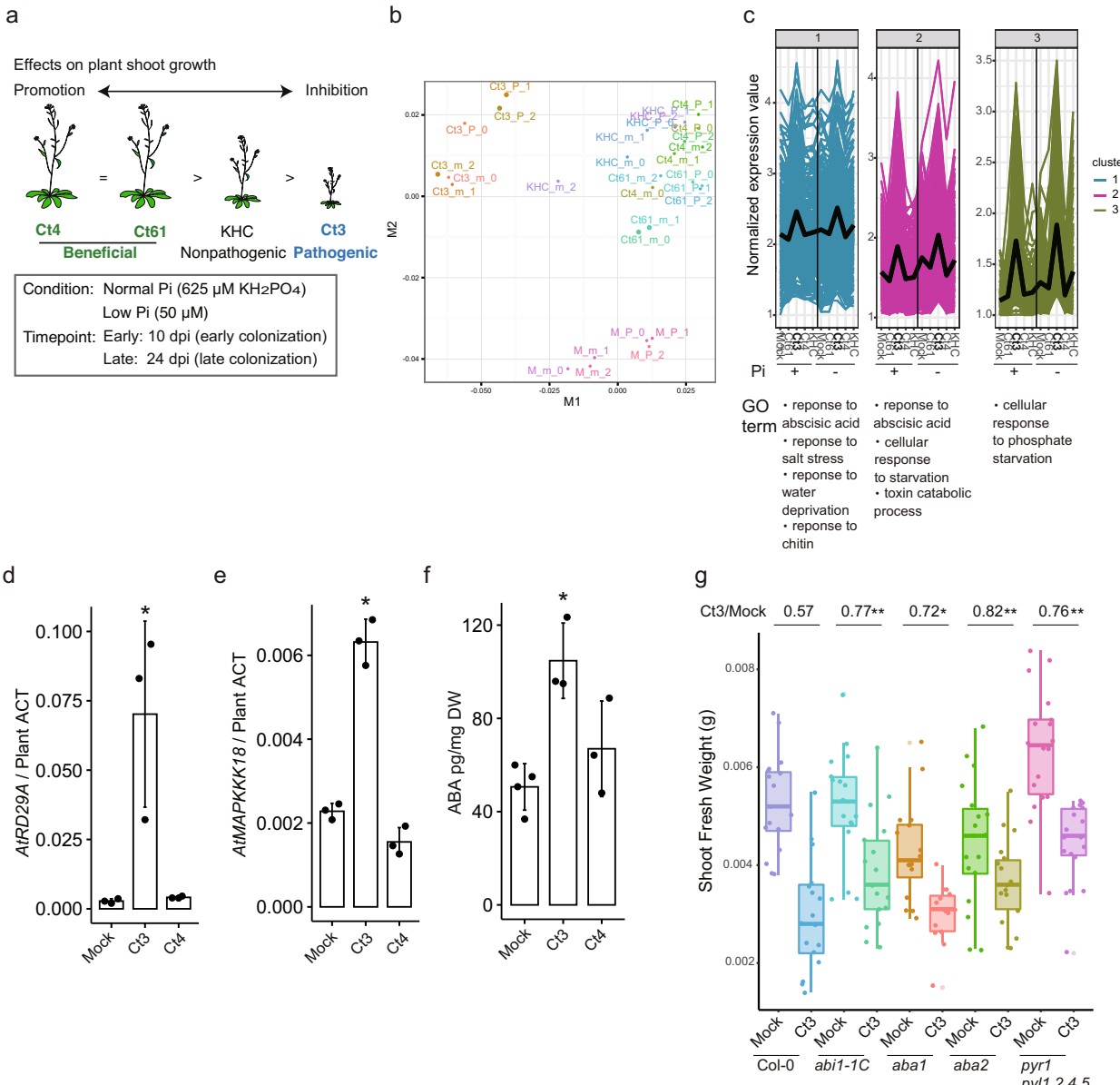

**Fig. 3 | Activation of the host ABA responses during Ct3 pathogenesis.**
**a** Experimental setup for dual RNA-seq analysis of *A. thaliana* and Ct transcriptomes. **b** Multidimensional scaling (MDS) analysis chart generated from plant transcriptome profile during association with the indicated Ct and KHC strains under normal (P) and low Pi (m) conditions, with three biological replicates. **c** Expression patterns of *A. thaliana* genes differentially expressed during pathogenic Ct3 association compared with beneficial Ct associations (( | log2FC | >1, FDR < 0.05), indicated by cluster analysis divided by kmean (K = 3, PAM clustering with Jensen-Shannon distance). Genes listed in each cluster were subjected to GO analysis (The full gene list is available in Supplementary Data 3). **d**, **e** Expression of ABA-responsive *A. thaliana* genes (*AtRD29A* and *AtMAPKKK18*) at 10 dpi with Ct3 or Ct4 under low Pi (±SD, $p$ = 0.025352 (d), $p$ = 0.000267 (e), two tailed $t$-test) ($n$ = 3 biologically independent samples). **f** Root ABA content following fungal inoculation under low Pi (Mock: $n$ = 4, Ct3: $n$ = 3, Ct4: $n$ = 3 biologically independent samples). Asterisks indicate significantly different means compared to the mock controls (±SD, $p$ = 0.002651, two tailed $t$-test). **g** Shoot fresh weight at 24 dpi with Ct3 under low Pi conditions in the indicated plant genotypes (Col-0_Mock: $n$ = 17, Col-0_Ct3: $n$ = 17, *abi1-1C*_Mock: $n$ = 17, *abi1-1C*_Ct3: $n$ = 17, *aba1*_Mock: $n$ = 16, *aba1*_Ct3: $n$ = 14, *aba2*_Mock: $n$ = 18, *aba2*_Ct3: $n$ = 17, *pyr1 pyl1 2 4 5*_Mock: $n$ = 18, *pyr1 pyl1 2 4 5*_Ct3: $n$ = 18 biologically independent samples). The median values are described within each boxplot. Asterisks indicate significantly different fresh weight ratio (Ct3/Mock) compared with Col-0 (*$p$ < 0.1, **$p$ < 0.05, two tailed $t$-test (Col-0 vs. *abi1-1C*: $p$ = 0.047, Col-0 vs. *aba1*: $p$ = 0.08, Col-0 vs. *aba2*: $p$ = 0.021, Col-0 vs. *pyr1 pyl1 2 4 5*: $p$ = 0.033)).

conservation of their cluster across these Ct genomes. Their encoded amino acid sequences were nearly identical (particularly in Ct4; Supplementary Fig. 4o: see below). These genes were silenced at 24 dpi when plant growth inhibited by Ct3 was alleviated (Supplementary Fig. 2g; Supplementary Table 4). Furthermore, pathogenic *C. incanum* displayed an upregulation of *CiABA3* and *CiBOT5* during root infection (Supplementary Fig. 4a). Therefore, co-activation of putative ABA and BOT biosynthesis genes was specifically associated with pathogenic lifestyles in the root-infecting *Colletotrichum* species.

Our comparative genomic analysis indicated that putative ABA and BOT biosynthesis genes are conserved as a single cluster in beneficial and pathogenic Ct genomes but are separate in the *B. cinerea* genome (Fig. 4b). Furthermore, these genes, in particular of the BOT cluster genes, show very high degrees of sequence conservation between *Colletotrichum* spp. (class Sordariomycetes) and *B. cinerea* (class Leotiomycetes). Such sequence conservation was unexpected, given the considerable phylogenetic distance between the two fungal lineages, which diverged approximately 261.6 million years ago[21].

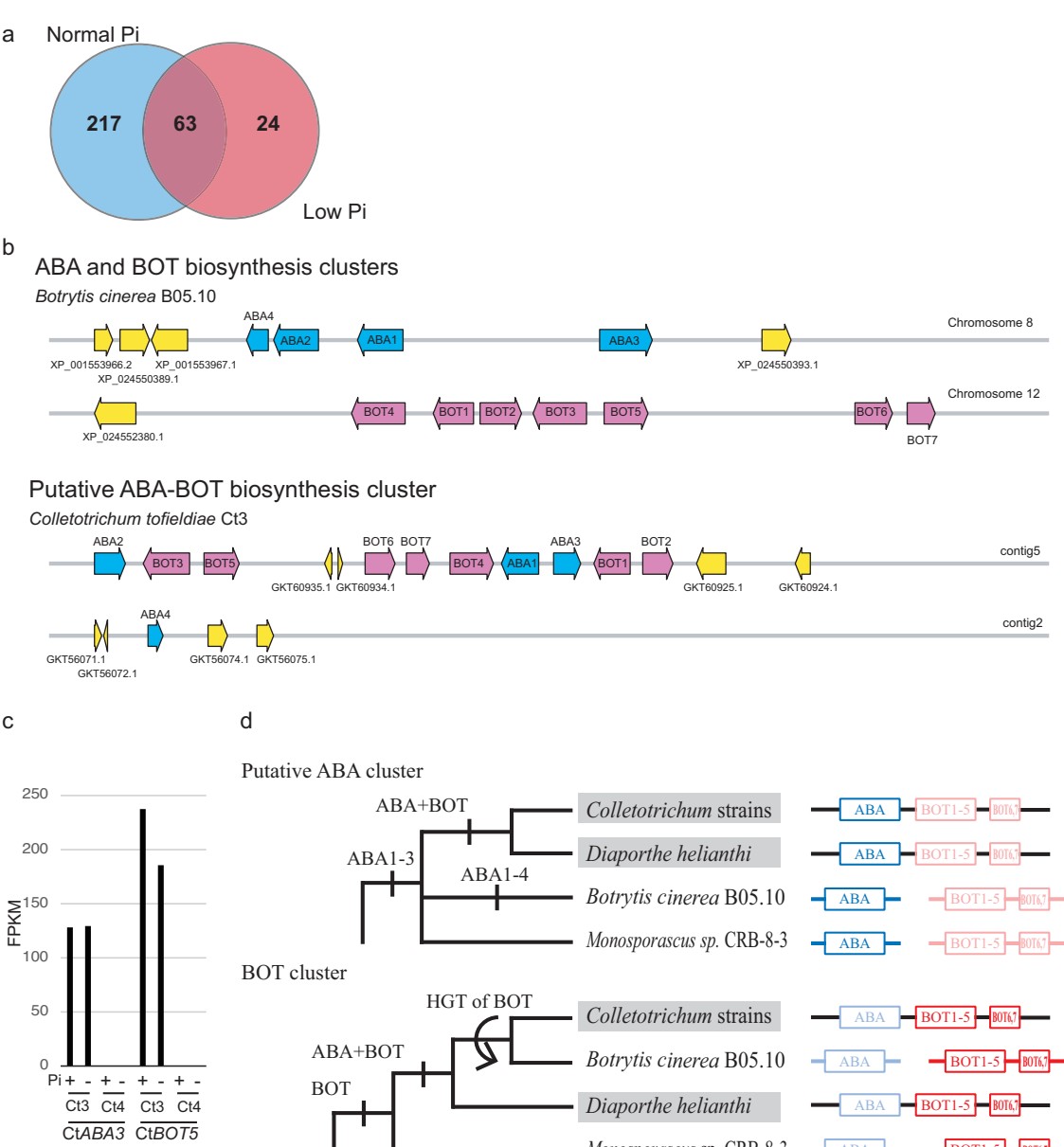

**Fig. 4 | Expression and distribution of the putative ABA and BOT biosynthesis-related genes in Ct are associated with fungal pathogenesis in Ct. a** Venn diagram showing Ct3 genes differentially upregulated in early colonization (10 dpi) compared with late colonization (24 dpi) phases (|log2FC|>1, FDR < 0.05, Supplementary Data 4). Sixty-three Ct3 genes were significantly upregulated during root colonization at 10 dpi in both Pi conditions. putative ABA and BOT biosynthesis genes were included in the Venn diagram. **b** Genomic structures of Ct3 putative ABA-BOT and *B. cinerea* ABA and BOT biosynthesis gene clusters. The blue and red arrows represent the genes related to ABA and BOT biosynthesis,

respectively. Yellow arrows represent other genes. **c** *CtABA3* and *CtBOT5* mRNA levels in Ct3 and Ct4 at 10 dpi, shown with the FPKM values being derived from RNA sequencing analysis. **d** Phylogenetic relationship and genomic structures of putative ABA and BOT gene clusters. All four genera carrying putative ABA and BOT gene clusters in their genomes were used for comparison. Organisms harboring adjacent putative ABA and BOT clusters (ABA + BOT (ABA-BOT) clusters) in their genomes are labeled with gray backgrounds. Phylogenetic analyses of these genes in the clusters are shown in Supplementary Fig. 4b–l.

To assess the evolutionary origin(s) of putative ABA and BOT biosynthesis genes, we generated molecular phylogenetic trees for these genes from the *Colletotrichum* strains (Ct3, Ct4, *C. liriopes*, *C. spaethianum*[34], belonging to the *spaethianum* clade), and from the GenBank non-redundant protein sequences (NR) database (Fig. 4d, Supplementary Data 6 and 7). Phylogenetic analyses showed (1) that putative ABA and BOT cluster genes were distributed across distantly related plant-associated fungi in a manner different from their phylogenic relationships (Supplementary Fig. 4b–m), (2) that putative ABA and BOT cluster genes each formed monophyletic groups (Supplementary Fig. 4b–m), (3) that genes in the putative ABA or BOT cluster each have similar evolutionary histories (Supplementary Fig. 4b–m),

but (4) that the putative ABA and BOT clusters have different evolutionary histories, indicated by their divergence in molecular phylogenetic trees (Fig. 4d and Supplementary Fig. 4b–m). The substantial differences between species and gene trees suggest horizontal gene transfers (HGTs) distributing the putative ABA and BOT biosynthesis gene clusters among these plant-associated fungi. Our comparative genomic analysis revealed that *Diaporthe helianthi*, a plant pathogen, also possesses a single putative ABA-BOT biosynthesis gene cluster of high synteny when compared with Ct (Supplementary Fig. 4n). On an assumption that the putative ABA-BOT cluster originated only once according to the principle of parsimony, the results suggest a HGT of the BOT cluster to *Botrytis*, (Fig. 4d and Supplementary Fig. 4m, n).

Given that the putative ABA and BOT clusters have likely arisen multiple times in different phytopathogenic fungi, it is conceivable that the acquisition of putative ABA and BOT biosynthesis gene cluster(s) contributes to pathogenesis evolution in plant-associated fungi.

Gene composition in putative ABA and BOT biosynthesis clusters was slightly different across species. The putative ABA biosynthesis cluster consists of *ABA1-ABA3* in, e.g., Ct, *C. incanum*, and *D. helianthi* or *ABA1-ABA4* in, e.g., *B. cinerea* (Fig. 4d, Supplementary Fig. 4b–e, and 4m, n). *CtABA4* was separately located in Ct genomes (Fig. 4b), and unlike *Ct3ABA1-Ct3ABA3*, its expression was not detected in Ct3 during root colonization (Supplementary Table 4), suggesting that *Ct3ABA4* is dispensable for Ct3 root colonization or pathogenesis in Ct3. The BOT gene cluster typically consists of *BOT1-BOT7*, where *BOT6* and *BOT7* correspond to Zn_clus (Zn$_2$Cys$_6$ transcription factor) and Adh_short (retinol dehydrogenase 8) genes, respectively, named in *B. cinerea*[33]. *BOT6* and *BOT7* show similar phylogenetic distributions with the other five BOT genes within *Colletotrichum* and *Botrytis* strains, indicating functional coupling of these genes in these fungal lineages (Supplementary Fig. 4f–n). We confirmed that *Ct3BOT6* and *Ct3BOT7* were strongly expressed during Ct3 pathogenesis (Supplementary Table 4).

### Ct3 produces intermediate metabolites of BOT biosynthesis during culture growth

Our phylogenetic analyses revealed a previously unsuspected diversity in the repertoires of biosynthetic genes for putative ABA and BOT among plant-associated fungi. We examined whether the biosynthetic genes in Ct3 mediate the production of these metabolites as predicted, like in *B. cinerea*. We generated fungal knockout mutants for *Ct3ABA2*, *Ct3ABA3*, and *Ct3BOT5* via homologous recombination. We then profiled metabolites related to BOT biosynthesis in the WT, Ct3Δ*aba2*, Ct3Δ*aba3*, and Ct3Δ*bot5* strains when cultivated in three distinct nutrient media (Mathurs, MPY, rice-based media). This detected two previously reported intermediate metabolites in BOT biosynthesis, 4β-acetoxyprobotryan-9β-ol (**9**) and 4β-acetoxyprobotryane-9β,15α-diol (**10**)[35,36], previously reported in *B. cinerea* in Ct3 (Fig. 5; Supplementary Fig. 5), specifically when cultivated in rice-based media (Fig. 5). Importantly, the accumulation of these metabolites was abolished in Ct3Δ*bot5*, disrupted at a putative catalyzing step upstream of their production, whereas it was unaffected in Ct3Δ*aba* (Fig. 5). The results indicate that the *Ct3BOT* contributes to the biosynthesis of BOT (**6**). In contrast, ABA (**S1**) or related precursor metabolites (**S3, S5**) were not detected under the conditions tested (Supplementary Fig. 6), implying a separation between ABA and BOT gene functioning, despite their co-clustering in the fungal genome and transcriptional co-regulation during root colonization. Alternatively, this may imply that the gene cluster is involved in the production of a metabolite or metabolites other than ABA.

### Genetic disruption of putative fungal ABA and BOT biosynthesis genes leads to switching from pathogenic to beneficial lifestyles in Ct3

We next tested whether putative fungal ABA and BOT biosynthesis genes are required for Ct3 pathogenesis. Ct3Δ*aba2*, Ct3Δ*aba3*, Ct3Δ*bot1*, Ct3Δ*bot3*, and Ct3Δ*bot5* mutant fungi showed WT-like growth and spore formation on nutrient-rich media and WT-like hyphal growth on glass slides (Supplementary Fig. 7a–c). Ct3Δ*aba2*, Ct3Δ*aba3*, and Ct3Δ*bot5* fungi showed WT-like growth on normal or low Pi media under our conditions (Fig. 6a), suggesting that these genes are dispensable for culture growth in Ct3.

We next examined a possible role for these fungal genes in the induction of host ABA responses after root inoculation. Compared to WT fungi, ABA-dependent *AtMAPKKK18* induction was reduced in the roots inoculated with Ct3Δ*aba2* and Ct3Δ*aba3* fungi, suggesting that the fungal ABA biosynthesis genes are required for the host ABA

responses (Fig. 6b). Interestingly, *AtMAPKKK18* induction was also absent in the roots inoculated with Ct3Δ*bot* fungi (Fig. 6b), suggesting a role for *Ct3BOT* genes in activating plant ABA responses. Furthermore, root fungal biomass was dramatically reduced for Ct3Δ*aba* and Ct3Δ*bot* fungi compared with WT fungi (Fig. 6c and Supplementary Fig. 7d-e). indicating that these fungal genes contribute to root infection of Ct3. Since Ct3 but not Ct4 increases fungal growth in the presence of sucrose (Supplementary Fig. 2h), we tested whether Ct3 root infection influences sucrose accumulation in the host. Ct3 colonization resulted in the accumulation of sucrose in roots, which was found to be dependent on the *Ct3ABA2* and *Ct3BOT5* genes (Supplementary Fig. 7f). These observations suggest that activation of the putative ABA-BOT cluster promotes sugar accumulation, which may turn, be exploited by Ct3 for extensive fungal growth in roots.

Plant growth inhibition was also reduced when inoculated with Ct3Δ*aba2* and Ct3Δ*aba3* fungi compared with WT fungi (Fig. 6d and Supplementary Fig. 7g, h). Ct3 virulence was, however, restored when ABA was exogenously applied (Supplementary Fig. 7i). *Ct3ABA3*-mediated plant growth inhibition was reduced in the host mutants defective in ABA signaling or biosynthesis (Fig. 6d). The results suggest that biosynthesis of ABA by Ct3 requires the plant ABA core pathway to promote fungal pathogenesis. Notably, exogenous ABA application also suppressed both plant growth and Ct4-mediated PGP under low Pi conditions (Supplementary Fig. 7j). The results are consistent with fungal-derived ABA playing a role in shifting from a beneficial to a pathogenic lifestyle in Ct. Remarkably, Ct3Δ*bot5* fungi did not inhibit but rather promoted plant growth under low Pi, reminiscent of Ct4 (Fig. 6e and Supplementary Fig. 7k). The mutation effects were abolished when *Ct3BOT5* was introduced back into the Ct3Δ*bot5* (Supplementary Fig. 7l). The results suggest that BOT biosynthesis is required for Ct3 virulence, and its disruption even renders Ct3 beneficial for the host.

### Ct3 shifts between pathogenic and beneficial lifestyles in a manner dependent on the temperature and host *AtPHR1/AtPHL1*

We noticed that Ct3 inoculation promoted plant growth under low Pi at 26 °C (Fig. 7a). This was accompanied by a decrease in fungal root colonization at 26 °C compared to 22 °C (Fig. 7b). Plant growth promotion and fungal colonization at high temperatures were unaffected for Ct3Δ*aba3* or Ct3Δ*bot5* fungi (Fig. 7a), consistent with a great decrease in *Ct3ABA3* and *Ct3BOT5* expression during root colonization at 26 °C (Fig. 7c). Limitation of Ct3 root colonization at high temperatures was thus associated with low expression of the putative fungal ABA-BOT cluster genes. However, in *phr1phl1* plants, Ct3 inhibited plant growth even at 26 °C in a manner dependent on the putative ABA-BOT cluster (Fig. 7a), suggesting that the suppression of fungal pathogenesis via the putative ABA-BOT at high temperatures requires the host *AtPHR1/AtPHL1*. Consistently, Ct3Δ*aba3* and Ct3Δ*bot5* fungi both promoted plant growth even in *phr1 phl1* plants at 26 °C (Fig. 7a). This also suggests that *AtPHR1/AtPHL1* are not required for PGP per se at high temperatures when the putative fungal ABA-BOT cluster is disrupted. Non-inoculated WT and *phr1 phl1* plants were indistinguishable in shoot growth at 26 °C even under low Pi, at least in our settings (Fig. 7a). These results indicate a critical role for the putative ABA–BOT cluster and its negative regulation by host *AtPHR1/AtPHL1*, in the pathogen-mutualist transition of Ct3.

### Host nitrogen and iron uptake genes are targeted by the putative fungal ABA-BOT cluster

*CtABA1, CtABA2, CtBOT1, CtBOT3*, and *CtBOT4*, all annotated to encode P450, constitute a large monophyletic group (Fig. 8a and Supplementary Fig. 8a). The simultaneous expression of these genes in conjunction with other Ct3 genes, suggests their interdependent regulation. Initially, we examined whether the expression of the 92 Ct3

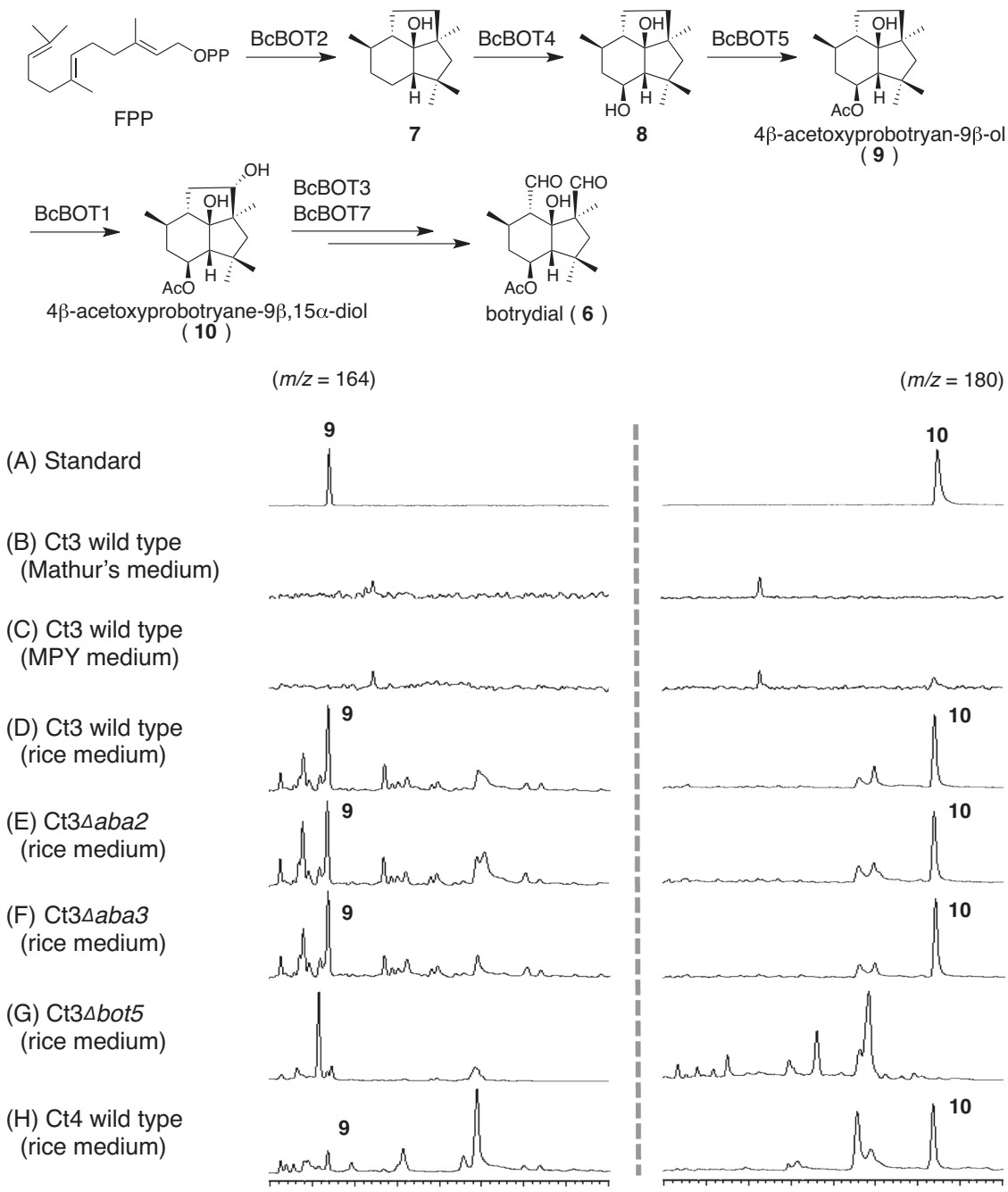

**Fig. 5 | Ct3 produces intermediate metabolites of BOT biosynthesis.** GC–MS profiles for 9 and 10 were obtained in the indicated Ct3 genotypes under the indicated medium conditions.

genes, which were co-expressed with putative ABA and BOT genes during pathogenesis, was altered in the Ct3Δaba2, Ct3Δaba3, and Ct3Δbot5 mutant strains in comparison to the wild type (WT) (Supplementary Data 8). Our findings indicate that the expression levels of these Ct3 genes remained largely unaffected in the mutant strains (Supplementary Data 9), implying a separate regulation of putative ABA and BOT genes from the other genes.

The requirements of putative ABA and BOT genes for activating the plant core ABA pathway during Ct3 infection implied the existence of a common host target(s) for the two fungal biosynthesis pathways. To gain insight into the host pathways affected by the putative fungal ABA-BOT cluster, we next assembled *A. thaliana* genes specifically induced or repressed by pathogenic Ct3, but not beneficial Ct4, in a

manner dependent on the putative Ct3 ABA-BOT cluster. We examined root transcriptome at 10 dpi with WT, Ct3Δaba2, Ct3Δaba3, and Ct3Δbot5 of Ct3, and with WT Ct4 under low Pi, where the *Ct3ABA/Ct3BOT* expression status greatly influences Ct3 lifestyles (Supplementary Data 10).

Plant responses were similar among Ct3Δaba, Ct3Δbot, and Ct4 (Supplementary Fig. 8b), consistent with their PGP effects under low Pi (Fig. 6e). Plant DEGs following inoculation with WT Ct3, compared with Ct3Δaba2, Ct3Δaba3, and Ct3Δbot5 mutants and WT Ct4, included 288 Ct3-induced genes and 375 Ct3-repressed genes (Fig. 8b–c and Supplementary Data 11 and 12). Ct3-induced genes were overrepresented with the genes responsive to oxidative stress, chitin, ABA, and cellular response to phosphate starvation. DNA

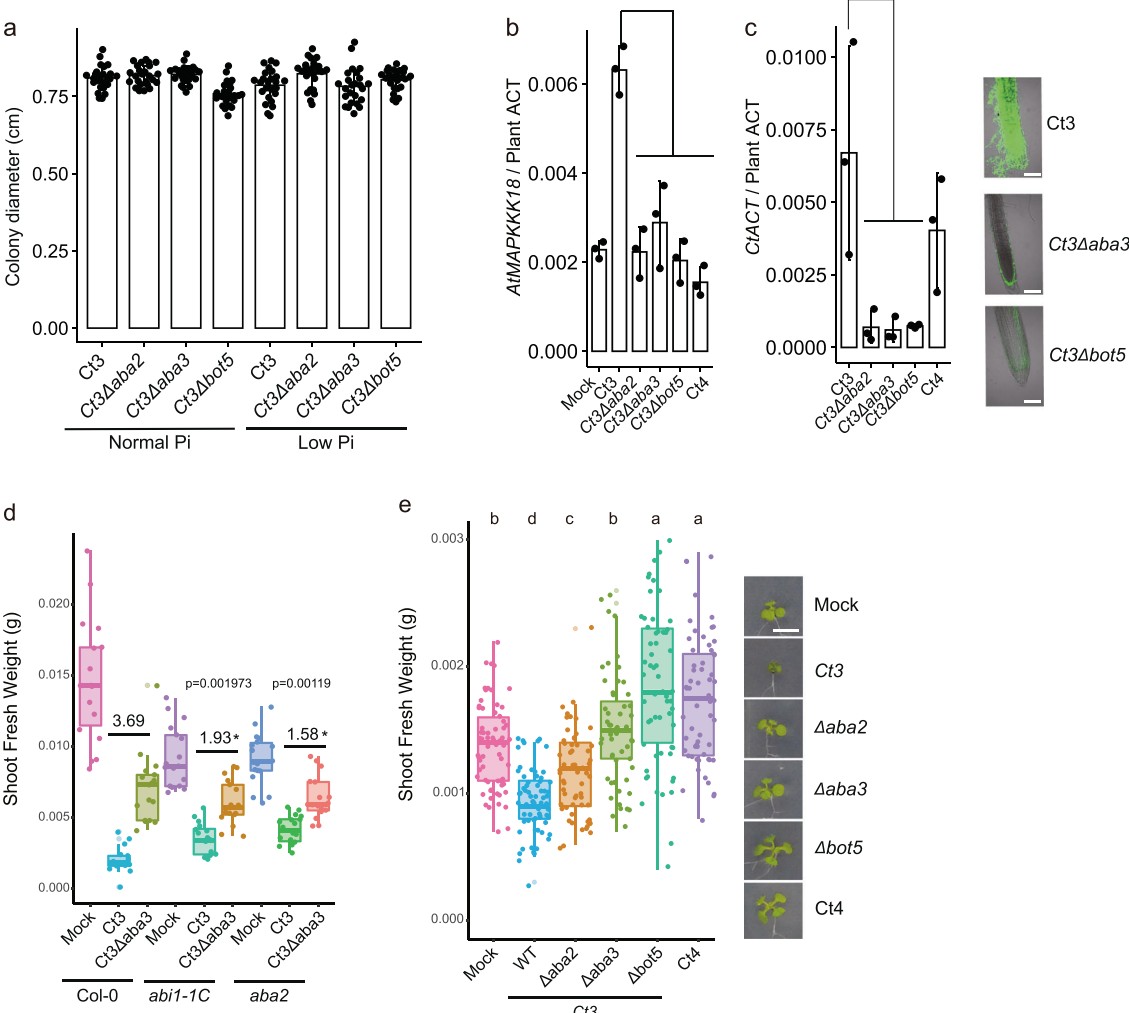

**Fig. 6 | Disruption of the putative fungal ABA and BOT biosynthesis genes results in a pathogen-to-mutalist lifestyle transition in Ct3 under low Pi.** **a** Fungal colony diameter under normal Pi and low Pi conditions. Bars represent ±SD ($n$ = 25 biologically independent samples). **b** *AtMAPKKK18* expression at 10 dpi with the indicated Ct genotypes under low Pi (±SD, $n$ = 3 biologically independent samples). **c** Fungal biomass indicated by *Colletotrichum* ACTIN relative abundance in *A. thaliana* roots at 10 dpi under low Pi (±SD, $n$ = 3 biologically independent samples). The photos represent the roots at 16 dpi with Ct3, $\Delta aba3$, and $\Delta bot5$. Ct hyphae were stained by WGA-Lectin (Bars = 100 μm). Asterisks indicate significantly different means compared with Ct3 ($n$ = 3 biologically independent samples,

$p < 0.05$, two-tailed $t$-test) in (**b**, **c**). **d**, **e** Shoot fresh weight at 24 dpi with the indicated fungi under normal Pi (**d**) and low Pi (**e**). The median values are described within each boxplot. Numerals indicate the $\Delta aba3$/Ct3 ratio, and asterisks indicate a significant difference compared to Col-0 ($p < 0.05$, two-tailed $t$-test) in (**d**) ($n$ = 17 or 18 (Col-0_Mock, $aba2$_Mock, and $aba2$_Ct3) biologically independent samples). Different letters indicate significantly different statistical groups in (**e**) (ANOVA, Tukey-HSD test, $p < 0.05$ (Mock: $n$ = 66, Ct3: $n$ = 68, Ct3$\Delta aba2$: $n$ = 61, Ct3$\Delta aba3$: $n$ = 56, Ct3$\Delta bot5$: $n$ = 60, Ct4: $n$ = 60 biologically independent samples)). Representative plant photos are shown (Bar = 1 cm).

motif analysis revealed DNA sequences bound by NAC transcription factors related to drought tolerance[37] being enriched within 1000 bp upstream of their transcription initiation sites (Supplementary Fig. 8c). Notably, the RNA-seq analyses also indicated that PSR gene activation at 10 dpi with Ct3 was dependent on the putative fungal ABA-BOT cluster (Fig. 8d). We further showed that expression of *AtAT4*, an AtPHR1-regulon during PSR, was significantly lowered in *abi1-1C* and *aba2* plants when colonized with Ct3 (Supplementary Fig. 8d), indicating a role for the host ABA pathway in Ct3 induction of host PSR genes. Interestingly, a large subset of GOs over-represented in the 375 Ct3-repressed plant genes were related to hosting nutrition, such as inorganic anion transport, cation homeostasis, and inorganic ion homeostasis (Supplementary Data 12 and Fig. 8c, e). Consistently, Ct3 root colonization changed host shoot nutrition status depending on putative ABA-BOT (Supplementary Fig. 8e). The results suggest that Ct3 root infection suppresses host nutrient uptake and impacts host mineral homeostasis.

To test the biological significance of this suppression, we examined whether the genes repressed by *Ct3ABA* and *Ct3BOT* genes and by host ABA pathways under low Pi contribute to plant growth (Fig. 8e). Ct3-repressible genes included *AtNRT1.1* and *AtNRT2.1*, two major transporters for nitrate uptake[38], and *AtFIT*, FER-like iron deficiency-induced transcription factor inducing Fe uptake genes[39]. Ct3 repression of *AtNRT1.1* and *AtNRT2.1* expression was alleviated in *aba2* and *abi1-1C* plants, respectively, pointing to its dependence on ABA (Supplementary Fig. 8f, g). Disruption of *AtNRT1.1*, *AtNRT2.1*, or *AtFIT* resulted in strong growth deficits or chlorosis under normal Pi in the absence of Ct3, whereas their disruption did not show significant effects under low Pi (Fig. 8f and Supplementary Fig. 8h, i). Although this has hampered assessing Ct3 infection phenotypes in these mutant plants, the results suggest that these nutrition-related genes are rate-limiting in plant growth under Pi sufficiency and that they define a host target for the fungal ABA-BOT cluster during Ct3-mediated plant growth inhibition.

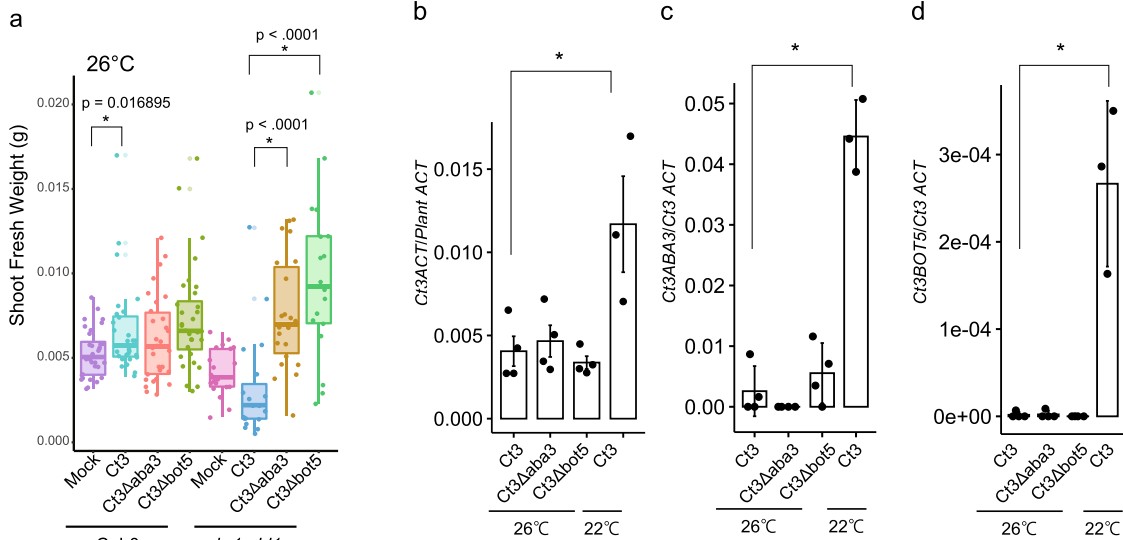

**Fig. 7 | Dynamic pathogenic-to-mutualistic lifestyle transitioning in Ct3 dependent on temperature and *AtPHR1*/*AtPHL1*. a** Shoot fresh weight at 24 dpi with the indicated fungi at 26 °C under low Pi. The median values are described within each boxplot. Asterisks indicate significantly different mean (Mock_Col-0: $n = 32$, Ct3_Col-0: $n = 28$, Ct3Δ*aba3*_Col-0: $n = 30$, Ct3Δ*bot5*_Col-0: $n = 32$, Mock *phr1 phl1*: $n = 20$, Ct3_*phr1 phl1*: $n = 19$, Ct3Δ*aba3 phr1 phl1*: $n = 22$, Ct3Δ*bot5 phr1 phl1*: $n = 18$ biologically independent samples, Mock vs. Ct3 (Col-0): $p < 0.05$, two tailed *t*-test). **b** Fungal biomass in roots at 24 dpi with the indicated fungi at 22 °C and 26 °C, indicated by RT-qPCR analysis (Ct *ACTIN*/Plant *ACTIN*) (±SD, Ct3: $n = 4$, Ct3Δ*aba3*: $n = 4$, Ct3Δ*bot5*: $n = 4$, Ct3_22: $n = 3$ biologically independent samples, $p = 0.034187$, two-tailed *t*-test). **c**, **d** *Ct3ABA3* or *Ct3BOT5* mRNA levels relative to *CtACTIN* in roots at 22 °C and 26 °C (±SD, Ct3: $n = 4$, Ct3Δ*aba3*: $n = 4$, Ct3Δ*bot5*: $n = 4$, Ct3_22: $n = 3$ biologically independent samples, $p = 0.000106$ (**c**), $p = 0.002197$ (**d**), two tailed *t*-test).

## Discussion

In this study, we obtain evidence in a Ct strain (Ct3) that a dynamic transition from pathogenic to mutualistic lifestyles during the root colonization is achieved by altering the expression of a single fungal secondary metabolism cluster (putative ABA-BOT) (Fig. 9). It has been well documented that the presence or absence of one genetic component(s) in the genome, e.g., an island, plasmid, or (mini) chromosome, is associated with the distinction between pathogenic and nonpathogenic (or potentially beneficial) lifestyles in different bacteria and fungi[40–42]. In contrast, our studies reveal that the beneficial and pathogenic Ct strains share a nearly identical putative ABA-BOT cluster in their genomes and that its expression status plays a critical role in the diversification of fungal lifestyles.

Our studies further indicate that altering Ct3 putative ABA-BOT cluster expression enables its lifestyle transition on the same host. The divergence between adapted and non-adapted (nonpathogenic) fungal pathogens was often attributed to transcriptional induction of virulence-related genes in the former[43,44]. Our findings extend this view that the loss of the putative ABA-BOT cluster not only results in the suppression of Ct3 pathogenesis but the expression of PGP function under low Pi, reminiscent of other beneficial Ct strains. This is also the case at high temperatures, where not only Ct3 virulence suppression but PGP is also achieved, likely through the host *AtPHR1*/*AtPHL1*-dependent suppression of putative ABA-BOT cluster expression. The results highlight an important role played by a secondary metabolism gene cluster in the plastic lifestyle transition of plant-infecting fungi and the convergence of environmental and host modulations on its transcriptional regulation. Notably, the investigated Ct strains, albeit classified into the Ct species through the widely accepted method for *Colletotrichum* species, harbor substantial genomic sequence variations. Although there are no sequence differences in the putative ABA-BOT region, there are subtle variations outside the putative ABA-BOT region between beneficial and pathogenic Ct strains, which might contribute to the divergence in the expression profiles of putative ABA-BOT genes and in fungal infection modes. Further studies are warranted to examine this hypothesis and the underlying mechanisms.

How does the Ct3 putative ABA-BOT cluster inhibit plant growth? Under our conditions, Ct3 and Ct4 are largely indistinguishable in root colonization levels (Fig. 1e–h, Fig. 6c), making it unlikely that Ct3 pathogenesis is caused by fungal overgrowth in the roots. It has been described that bacterial and fungal pathogens mobilize the host ABA pathway to suppress plant immunity, in particular, salicylic acid (SA)-based defenses[28]. However, our transcriptomic analyses on Ct3 wild-type and Ct3Δ*aba*/Ct3Δ*bot* fungi did not detect ABA/BOT-dependent alterations in plant induction of defense-related genes. Consistently, beneficial Ct61 promotes plant growth in the plants simultaneously disrupted with defense-related hormones, SA, jasmonate, ethylene, and defense regulator *AtPAD4*, as well as in the WT plants, without fungal overgrowth[3]. These data suggest that Ct3-mediated virulence is not expressed through ABA-SA antagonism. Rather, our transcriptome and genetic data suggest that Ct3 utilizes the putative ABA-BOT cluster to suppress the host genes required for the acquisition of different key nutrients, such as nitrogen, iron, and zinc, which are rate-limiting in plant growth, particularly when phosphate is sufficient. These results suggest that Ct3 virulence via putative ABA-BOT occurs at least in part through perturbation of host nutrition. Indeed, Ct3 putative ABA-BOT contributes to sucrose accumulation in the host roots (Supplementary Fig. 7f), which is likely effectively exploited by the fungus for increasing hyphal growth.

Host plants require *AtPHR1*/*AtPHL1* for suppression of fungal overgrowth in beneficial interactions with Ct61, a prerequisite for PGP[3]. This work further shows the critical role of *AtPHR1*/*AtPHL1* in restricting Ct3 pathogenesis. Consistently, PSR-related *AtPHR1*/*AtPHL1* regulons are induced during both pathogenic and beneficial interactions, albeit at different timings. In Ct3, putative ABA-BOT genes serve to accelerate the early induction of host PSR-related genes (at 10 dpi) even under Pi-sufficient conditions, likely through the host ABA core pathways. This may reflect a positive role for ABA in PSR under low Pi[45,46]. In contrast, beneficial Ct strains induce host PSR-related genes, specifically under low Pi, at a later phase (24 dpi), coincident with the appearance of PGP effects. Given the absence of *CtABA*/*CtBOT* gene expression in beneficial Ct strains, their PSR activation seems

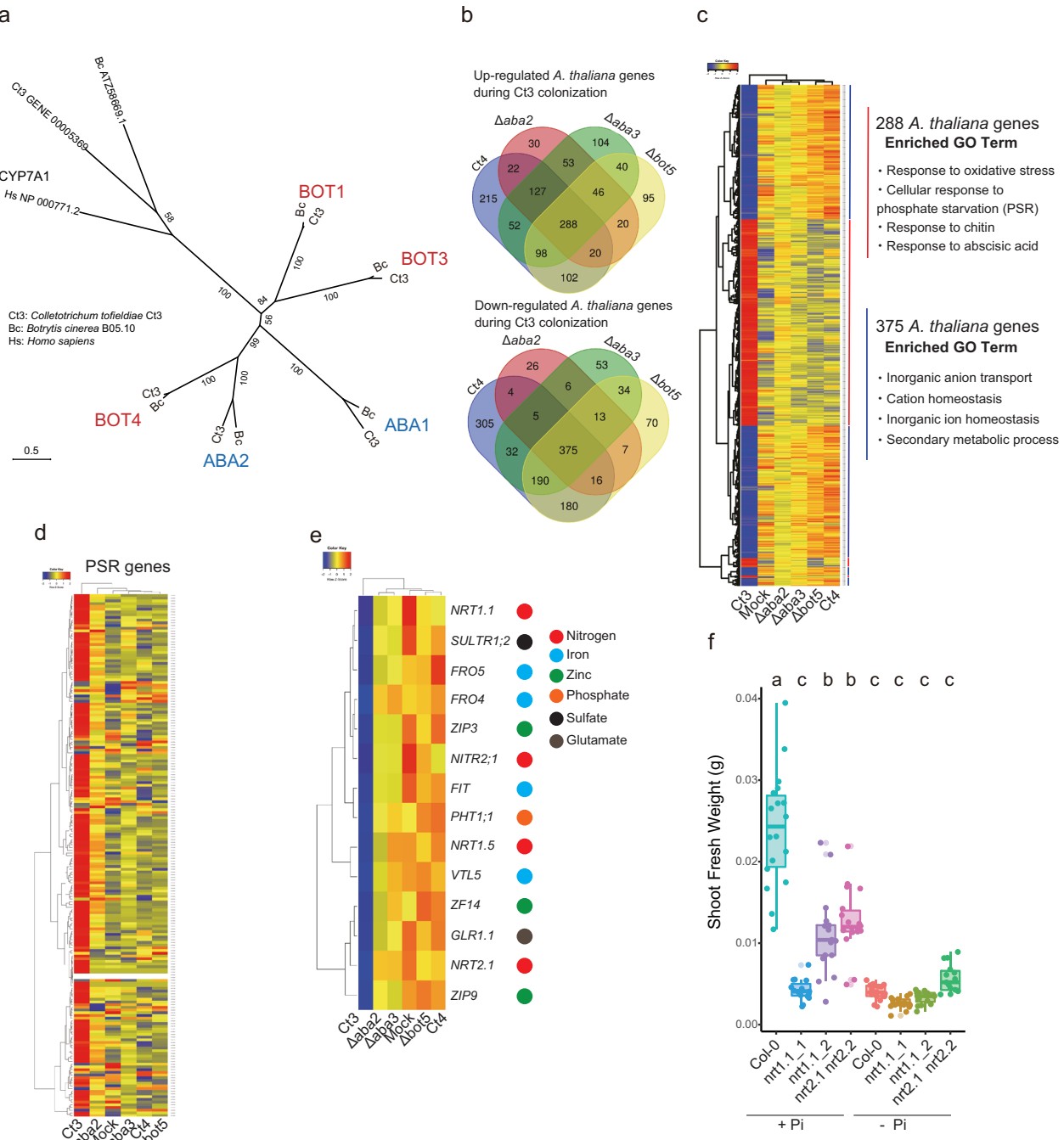

**Fig. 8 | Ct3ABA-BOT gene cluster suppresses nitrogen uptake genes while activating PSR genes in the host during Ct3 colonization. a** Maximum Likelihood tree of ABA and BOT genes in Cytochrome P450 family using IQ-TREE version 1.6.11. Ct3: *C. tofieldiae* Ct3; Bc: *B. cinerea* B05.10; Hs: *Homo sapiens*. Homologs to CYP7A1 were used as an outgroup. The phylogenetic relationship of the ABA and BOT genes and other 409 P450 genes in *C. tofieldiae*, *B. cinerea*, and *H. sapiens* is indicated in Supplementary Fig. 8a. Ultrabootstrap probability is shown on the branches. The scale bar represents substitutions per site. **b** Venn diagram representing significantly up (Upper)- or down (Down)-regulated *Arabidopsis* genes at 10 dpi with Ct3 compared with the other Ct genotypes (Ct4, Ct3Δ*aba2*, Ct3Δ*aba3*, and Ct3Δ*bot5*) (|log2FC|>1, FDR < 0.05) under low Pi. **c** Transcript profiling of 288 commonly up-regulated and 375 commonly down-regulated *Arabidopsis* genes in the roots following inoculation with all the examined Ct genotypes. Overrepresented (red to yellow) and underrepresented (yellow to blue) modules are depicted as log10 (fpkm + 1). The major enriched GOs of 288 or 375 genes IDs are

enlisted, respectively. Red and blue lines next to the gene IDs represent their upregulation or downregulation, respectively. **d** Hierarchical clustering of *A. thaliana* PSR-related 193 genes. Overrepresented (red to yellow) and underrepresented (yellow to blue) modules are depicted as log10 (fpkm + 1). **e** Hierarchical clustering of *A. thaliana* genes related to nutrient uptake and suppressed by Ct3 through the ABA-BOT cluster (FDR < 0.05). Overrepresented (red to yellow) and underrepresented (yellow to blue) modules are depicted as log10 (fpkm + 1). **f** Shoot fresh weight of the indicated *Arabidopsis* genotypes under normal (+) or low (−) Pi at 24 days (Col-0: *n* = 18, *nrt1.1_1*: *n* = 15, *nrt1.1_2*: *n* = 17, *nrt2.1 nrt2.2*: *n* = 18, Col-0_low Pi: *n* = 18, *nrt1.1_1* low Pi: *n* = 17, *nrt1.1_2* low Pi: *n* = 16, *nrt2.1 nrt2.2* low Pi: *n* = 15 biologically independent samples). The median values are described within each boxplot. The fresh weight of each plant was measured after 24 days of incubation. Different letters indicate significantly different statistical groups (ANOVA, Tukey-HSD test, *p* < 0.05).

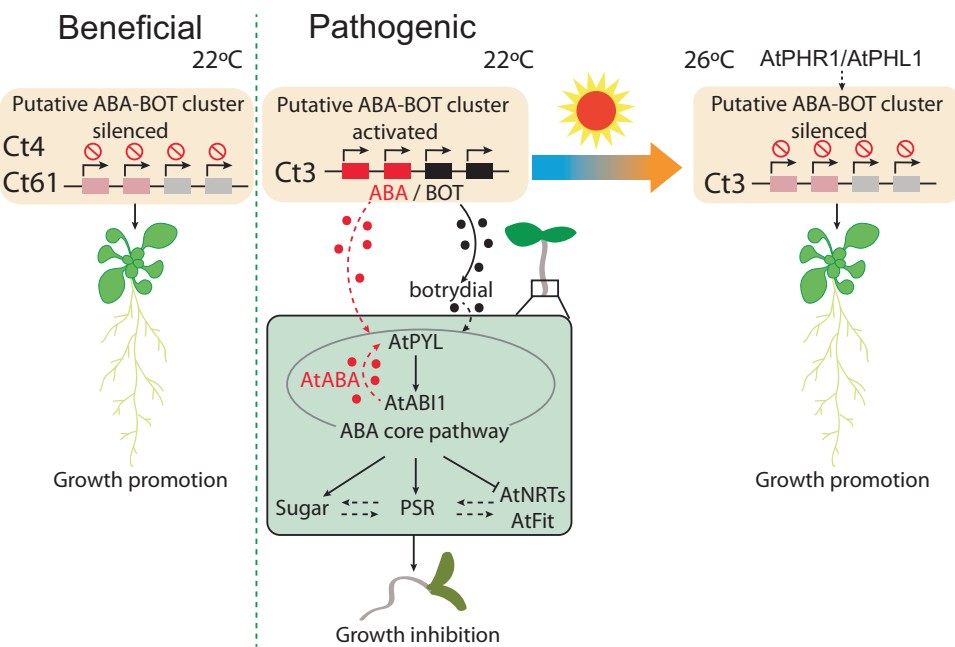

**Fig. 9 | A model depicting the pathogenic-mutualistic lifestyle transition in Ct3.** Unlike beneficial Ct strains (Ct4 and Ct61), Ct3 induces putative ABA-BOT gene expression during an initial phase of root colonization. This putatively leads to fungal production of ABA, its precursors, and/or related metabolites from the ABA biosynthetic genes as summarized previously[16] and botrydial production from the BOT biosynthetic genes as shown in Fig. 5. These metabolites, in turn, involve the host ABA core pathway, thereby leading to the suppression of nutrient uptake-related genes and the activation of PSR-related genes in *A. thaliana*. Perturbation of the host nutrition pathways is then associated with sucrose accumulation in the roots, which is likely to facilitate Ct3 hyphal growth. At 26 °C, Ct3 colonizes the roots with the putative ABA-BOT genes silenced through the host AtPHR1/AtPHL1 and promotes plant growth under low Pi.

independent of putative fungal ABA or BOT biosynthesis genes. Indeed, like Ct3, beneficial Ct strains efficiently colonize *A. thaliana* roots under low Pi without expressing *CtABA/CtBOT* genes (Figs. 1e, f and 6c). Therefore, separate fungal mechanisms, in terms of ABA dependence, are likely to confer host PSR activation between pathogenic Ct3 and beneficial Ct interactions. Conversely, *AtPHR1/AtPHL1* restricts fungal growth under low Pi through the suppression of separate fungal infection strategies between Ct3 and beneficial Ct strains, diverging in putative ABA-BOT dependence. Our evidence attributes *AtPHR1/AtPHL1*-mediated alleviation of Ct3 pathogenesis to the suppression of fungal putative ABA-BOT expression, under low Pi and at high temperatures. Plant transcriptome data not detecting differential defense gene regulation imply that *AtPHR1/AtPHL1*-mediated control of fungal growth is also distinct from their suppression of pattern-triggered immunity[10,11].

Interestingly, the loss of *AtPHR1/AtPHL1* allows Ct3Δ*aba3* and Ct3Δ*bot5* fungi to promote plant growth in *phr1 phl1* at elevated temperatures, which is not seen in the WT plants (Fig. 7a). This indicates that *AtPHR1/AtPHL1* are not required for the PGP per se conferred by Ct3 at high temperatures when the putative ABA-BOT cluster is disrupted. In addition to stimulating host ABA signaling, putative ABA-BOT may suppress the PGP function through promoting host Pi acquisition, which is conserved in Ct species, including Ct3 (suggested by Fig. 7a). This also seems to apply to pathogenic *C. incanum*, in which inherent putative ABA-BOT expression is associated with a massive decrease in phosphorus transfer to the host compared to beneficial Ct61[3]. Notably, however, in Ct3, this PGP function is likely suppressed by *AtPHR1/AtPHL1*, suggested by the absence of PGP in WT plants when inoculated with Ct3Δ*aba3*/Ct3Δ*bot5* fungi (Fig. 7a). It is conceivable that ABA-dependent activation of *AtPHR1/AtPHL1* regulons activated at 10 dpi with Ct3 contributes to the suppression of PGP, given the previously described negative role for *AtPHR1* in nitrate transporter expression, including *AtNRT1.1*, required for plant growth[47,48]. The mechanisms underlying the multifaceted functions of *AtPHR1/AtPHL1* in the complex host–fungus interactions require further studies.

Consistent with our molecular phylogenetic analysis that the beneficial strains are derived from the pathogenic ancestors (e.g., *C. incanum* or *C. liriopes*), the present evidence indicates that the acquisition and expression of putative ABA-BOT gene cluster (or putative ABA and BOT gene clusters) contribute to fungal pathogenesis, likely through the activation of host ABA biosynthesis and signaling during infection. It seems that putative ABA and BOT gene clusters also contribute to fungal pathogenesis in *Botrytis*, as shown for Ct since host ABA signaling is required for the infection and pathogenesis of the necrotrophic fungus[18,19]. Although definite proof remains to be obtained, putative ABA-BOT gene clusters are likely to confer virulence in phylogenetically diverse fungi beyond Ct species. The additional requirements for host ABA biosynthesis in fungal pathogenesis imply that these fungal genes alone are not sufficient for effective ABA biosynthesis and virulence. Future studies will be required to elucidate the precise mechanisms by which fungal and plant ABA pathways work in concert to promote fungal virulence.

## Methods

### Plant material and growth conditions

*Arabidopsis thaliana* Col-0, the *abi1-1C*[27], *aba1*[49], *aba2-12*[50], *pyr1-1 pyl1 pyl2 pyl4 pyl5*[51], *phf1*[52], *phr1 phl1*[9], *nrt1.1 (chl1-5*[53]), *nrt2.1nrt2.2*[54], and *fit1-2*[39] mutants (Col-0 background) and *nrt1.1_2* (Ler background) were used in this study. We also used 15 *A. thaliana* WT accessions as described in Supplementary Fig. 2. Seeds were surface sterilized with 70% ethanol for the 30 s, followed by 6% sodium hypochlorite with 0.01% Triton X. After being washed three times in sterilized water, seeds were placed in cold treatment at 4 °C for 24 h before sowing.

### Fungal inoculation assay

Sterilized *A. thaliana* seeds were sowed on half-strength MS agarose medium containing defined Pi concentrations. Ct spores (3 μL of $1 \times 10^4$/ml) were inoculated 3 cm below the sowed seeds. Plates were placed vertically in a plant growth chamber under a 10:12 h light: dark cycle at ~22 °C (±1 °C) (80 μmol/m²s) around 50% humidity unless

otherwise described. For 26 °C assays, plants were incubated in a plant growth chamber under a 10:12 h light: dark cycle at ~26 °C (80 μmol/m²s) during light and ~22 °C during dark. Effects on plant growth by fungal inoculation were evaluated by measuring shoot fresh weight, root length, or chlorophyll a and b from ~15 plants per experiment. Ct hyphae were mixed with soils for soil inoculation, and *B. rapa* seeds were incubated for 24 days on the soils before measuring shoot fresh weight (SFW).

## Plant growth conditions

Half-strength MS medium used in this study is based on the previous work[55], with minor modifications. Agar granulated (Difco) was initially used as an agar containing limited phosphate in this study. However, as the new batches of the agars caused unstable growth in mock-treated plants under low Pi, INA Agar BA-10 (INA food) with further limited phosphate was used as an alternative. pH was adjusted to 5.1. 55 ml of the mixture was then poured into square Petri plates (15 × 15 cm, Greiner). For Ct inoculation in soils, Ct spores (50 mL of $1 \times 10^4$/ml) or distilled water (mock treatment) were incubated on mixtures of autoclaved 180 g sawdust, 60 g rice bran, 60 g bran, and 180 ml water for 2 weeks, where mixtures with or without Ct hyphae were mixed with soils containing Kanuma soil, Exo sand, and black soil (15:42.5:42.5) with 3-5% weight. Before measuring SFW, *B. rapa* var. *perviridis* seeds ('Misaki', Sakata seeds) were then sowed in the soil and incubated for 24 days.

## Fungal growth assay in vitro

Agar plugs of ten-day-old fungal cultures grown on Mathur's media were transferred to new Mathur's plates. Alternatively, 3 μL of fungal spore suspension, comprising approximately $1 \times 10^4$/ml, was put onto standard Pi media (625 μM $KH_2PO_4$) and low Pi media (50 μM $KH_2PO_4$) without sucrose. Colony formation was determined three days after inoculation by measuring the colony radius from center to edge. Fungal spore counts were conducted as described previously[56]. For fungal growth on glass slides, fungal spores were placed on glass slides and incubated for 3 days.

## Leaf chlorophyll measurements

The method of chlorophyll quantification was adapted from the previous report[57]. Samples were prepared from ~100 mg of leaf tissue pooled from ~4 plants per sample and weighed. Chlorophyll was extracted by adding 800 μL chilled cold 100% acetone, and samples were shaken for 10 min until plant tissue was transparent. After the samples were diluted four times with 80% acetone, the absorbance of tissue-free chlorophyll extract was measured at 646 nm, 663 nm, and 750 nm with an Eppendorf Biospectrometer following the formula.

A: Chlorophyll a (μg/ml) $= 12.25 \times (OD_{663.6} - OD_{750}) - 2.85 \times (OD_{646.6} - OD_{750})$

B: Chlorophyll b (μg/ml) $= 20.31 \times (OD_{646.6} - OD_{750}) - 4.91 \times (OD_{663.6} - OD_{750})$

## Quantitative real-time PCR

cDNA was synthesized from 300 to 500 ng total RNA using the PrimeScript RT Master Mix (Takara) in a volume of 10 μL. We then amplified 3 μL of cDNA (10 ng/μL) in Power SYBR (Thermo) with 1.6 μM primers using the AriaMx real-time PCR system (Agilent) in a volume of 12 μL. Primers used in this study are listed in Supplementary Table 5.

## Fungal transformation via *Agrobacterium*-mediated transformation

For targeted gene replacement, we used the previously described method[3]. Briefly, to generate replacement mutants lacking Ct3 putative ABA biosynthesis genes (*ABA2* and *ABA3*) or BOT biosynthesis genes (*BOT1*, *BOT3*, and *BOT5*), we constructed a plasmid in which the Hyg resistance gene cassette was inserted between DNA sequences

flanking the target gene. Around 1.5-kb fragments of 5′ (5 F) and 3′ sequences (3 F) flanking the gene ORF were amplified from Ct3 genomic DNA by PrimeSTAR HS DNA polymerase (TaKaRa). The purified PCR products 5 F and 3 F were mixed with *Sal*I-treated pBIG4MRHrev for In-Fusion reactions. This generated plasmid was then introduced into *Agrobacterium* C58C1 strains by electroporation using the default setting for the Eppendorf Eporator. The transformed *Agrobacterium* strains ($OD_{600} = 0.4$) were mixed with Ct3 spores ($1 \times 10^7$) in a 1:1 ratio on a paper filter attached to incubation media containing 200 μM acetosyringone (BLD Pharm). Selection of hygromycin (150 μM)-resistant strains was conducted by transferring the paper filter to PDA media with Hyg, cefotaxime, and spectinomycin (all at 50 μg/mL) and incubated for two days before the paper was eliminated from the media. The resistant strains from the media were tested by PCR using primers targeting the outside sequence and a Hyg sequence to determine whether the Hyg resistance cassette successfully replaced the interesting genes. The PCR primers used are listed in Supplementary Table 5. The Ct3 and Ct4 lines constitutively expressing GFP were generated as previously described[3].

## Fluorescence microscopy

Inoculated *A. thaliana* roots were observed through confocal microscopy. To trace the colonization process of Ct3-GFP or Ct4-GFP in the roots, we utilized a confocal laser scanning microscope (Nikon C2 plus) equipped with filters optimized for visualization of GFP or mCherry to distinguish fungal hyphae and plant plasma membranes. Fungal lectin staining was performed using a confocal laser scanning microscope Olympus FV1000 with filter settings suitable for fluorescein (FITC), i.e., blue excitation light. WGA-FITC conjugate (Sigma) was used to specify the fungal cell walls. Fungal-inoculated roots were incubated overnight in a 1:3 mixture of chloroform and ethanol. Then, the roots were transferred to chloral hydrate (2 g/mL in water) and incubated for 2 h. After PBS washed roots, the roots were incubated in wheat germ agglutinin (WGA) conjugated to a FITC solution (5 μg/ml; Sigma) for 1 h at room temperature.

## Genome sequencing and assembly

DNA extraction, genome sequencing, and assembly were conducted as described[34]. De novo assembly of Ct4 genomes was conducted by FALCON-integrate (v. 1.8.18). Illumina HiSeq for 100 bp paired-end short reads against Ct4 genomes was also conducted by Macrogen (TrueSeq DNA PCR-Free kit), resulting in 59 million filtered reads. The filtered sequence reads were used for error collection by mapping Hiseq reads to the PacBio assembly using Pilon (v. 1.21).

## Syntenies of Ct3, Ct4, and Ct61 genomes

Syntenies of Ct3, Ct4, and Ct61 genomes were detected as locally collinear blocks (LCB) partly by performing whole-genome alignment in the Mauve program v1.1.1[58]. The location of LCBs and synteny correlations were visualized as circos diagram by package circlize v.0.4.3[59] in R.

## Phylogenetic tree analyses for fungal classification

The maximum-likelihood phylogenetic tree was reconstructed using 6 fungal marker nucleotide sequences of 34 strains in 12 species of the genus *Colletotrichum*. The 6 sequences used were the nuclear 5.8 S ribosomal RNA gene with the two flanking internal transcribed spacers (ITS), a 200-bp intron of the glyceraldehyde-3-phosphate dehydrogenase gene (*GAPDH*), partial sequences of the actin (*ACT*), chitin synthase 1 (*CHS-1*), beta-tubulin (*TUB2*), histone 3 (*HIS3*) genes, as used previously to identify the species and species complexes of the genus *Colletotrichum*[23,25]. Sequence information for all the strains was retrieved using BLAST. The sequences of *C. incanum* MAFF 238704 and *C. liriopes* MAFF 242679 in GenBank were also used. Nucleotide sequences were aligned using MAFFT[60] and concatenated manually.

The maximum-likelihood phylogenetic tree was reconstructed using IQ-TREE (v.2.1.2) with the options *-m MFP -T AUTO -B 1000*[61].

The maximum-likelihood phylogenetic tree was also reconstructed using single-copy orthologs of all genomes of the genus *Colletotrichum* in GenBank, those obtained in this study, and that of *B. cinerea* B5.10 as an outgroup. The predicted proteins of the 71 fungal strains were clustered into single-copy orthologous groups using Orthofinder v.2.5.5[62] and Mirlo (https://github.com/mthon/mirlo). Amino-acid sequences of the 1,509 single-copy orthologous groups were aligned using MAFFT with default settings[60]. The maximum-likelihood tree was reconstructed using IQ-TREE (v.2.0.3) with the options *-m MFP -B 1000*[61].

### Construction of phylogenetic trees of ABA and BOT biosynthesis genes

For this analysis, we re-annotated putative ABA and BOT biosynthesis Ct (Ct3, Ct4, and Ct61) genes based on both genome assembly and RNA-seq expression data for each gene. Sequences similar to ABA and BOT genes of *Colletotrichum* strains, *B. cinerea*, and some other strains were collected with BLASTP search using Biopython (v.1.7.6; *E*-value: 1e-4 or 1e-5) against newly analyzed genomes in this study, *Homo sapiens* genome assembly GRCh38.p13, and GenBank NR database on February/March 2020. Those containing large indels or that were highly diverged were omitted from the data set of sequences (Supplementary Table 3). Sequences were aligned with MAFFT using the E-INS-i strategy (v.7.273)[60]. Phylogenetic analysis was conducted with IQ-TREE (v.1.6.11; *-m MFP -nt AUTO -b 100*)[61]. Microsynteny of ABA and BOT genes was analyzed with GenomeMatcher (v.3.00)[63]. The fungal species tree in Fig. 4m is followed by[64].

### RNA extraction and RNA-sequencing analyses

Total RNA was extracted from the whole root compartments using the NucleoSpin RNA Plant kit (Macherey-Nagel). RNA samples (1 μg each) were then sent to BGI for quality assurance, library preparation, and subsequent sequencing using their default process. The generated libraries were sequenced by HiSeq-illumina, resulting in approximately 40 million reads per sample, paired-end 150 bp (Figs. 3 and 4) or BGIseq (Fig. 8), resulting in about 40 million reads per sample, paired-end 100 bp. Tophat2 with default settings, except that average fragment size was specified[65], was used for sequence mapping based on the *A. thaliana* genome (TAIR10) or fungal genomes. The generated BAM files from the Tophat2 platform were used to analyze DEGs by cuffdiff with default settings[66]. The following statistical analysis and data visualization were conducted in R Bioconductor packages, such as Cummerbund, gplots, and ggplot2[66,67]. As judged by the Cummerbund platform, DEGs were identified using an FDR threshold of < 0.05. Heatmaps representing gene expression profiles were generated with the R package heatmap.2 in gplots. GO analysis for *A. thaliana* genes was conducted by AgriGO v. 2 (FDR < 0.05[68]). Motif analysis was conducted by CentriMo[69] with the default setting using 1000 bp upstream of the target *A. thaliana* genes extracted via TAIR Bulk Data Retrieval.

### Profiling fungal metabolites during liquid culture growth

The mycelia of Ct were inoculated into 1 mL of both MPY and Mathur's medium in a 5 mL test tube. The cultures were incubated at 25 °C with a constant rotational speed of 200 rpm for a period of 5 days. Following extraction with 2 mL of acetone, the extracts were evaporated. Subsequently, 1M-HCl was added to the mixture, and the resulting mixture was extracted with ethyl acetate. The crude extracts were then dissolved in methanol.

### For botrydial intermediates

After filtration of the crude extracts, the filtrates were directly analyzed by a GC-MS equipped with a Beta DEX™ 120 fused silica capillary

column (0.25 mm × 30 m, 0.25 μm film thickness; SUPELCO) in the following conditions (Method A); 100 °C for 3 min, 100-230 °C (rate: 14 °C/min), 230 °C for 5 min at a flow rate of 1.2 mL/min (helium carrier gas). The followings are summary of other setting conditions; Column Oven Temp.: 100 °C, Injection Temp.: 230 °C, Pressure: 89.4 kPa, Total Flow: 28.1 mL/min, Column Flow: 1.2 mL/min, Linear Velocity: 40.7 cm/sec, Purge Flow: 3.0 mL/min, Split Ratio: 20.0, Ion Source Temp.: 200 °C, Interface Temp.: 230 °C, Solvent Cut Time: 2.5 min, Micro Scan Width: 0 u, and Mass range: 50-500. GCMSsolution software (version 4.45) was used for data acquisition and data analysis.

### For ABA intermediates

After filtration of the crude extracts, the filtrates were directly analyzed by a UPLC-MS equipped with ACQUITY UPLC BEH C18 (2.1 × 50 mm) in the following conditions (Method B); 0-1 min = 10% B, 1-3 min = 10-95% B, 3-5 min = 95% B (A: $H_2O$ + 0.1% of formic acid, B: $CH_3CN$ + 0.1% of formic acid) at a flow rate of 0.7 mL/min; column temp. 40 °C. The followings are a summary of other setting conditions; $\lambda$ Range: 210 nm-400 nm, Resolution: 1.2 nm, Sampling Rate: 20 points/s, Filter Time Constant: 0.2 s, Exposure Time: auto. MassLynx V4.1 was used for data acquisition and data analysis.

### Profiling fungal metabolites during solid medium incubation

Mycelia of Ct was inoculated into a solid medium containing polished rice (1 g) in a 5 mL test tube. The culture was incubated at 25 °C for 5 days. After extraction with Ethyl acetate, the extract was concentrated in vacuo to afford crude extracts.

### For botrydial intermediates

After filtration of the crude extracts, the filtrates were directly analyzed by a GC-MS in Method A (see above).

### For ABA intermediates

The crude extracts were evaporated, and the residues were then dissolved in MeOH. After filtration, the filtrates were directly analyzed by a UPLC-MS in Method B (see above).

### Isolation of the biosynthetic intermediates

The ABA biosynthetic intermediates were isolated from the transformants constructed in our previous studies[17,70]. The BOT biosynthetic intermediates, **9** and **10**, were isolated from the *A. oryzae* transformants possessing either *BcBOT2/4/5* or *BcBOT1/2/3/4/5*, which were constructed by incorporating those biosynthetic genes into *A. oryzae* applying an established hot spot-knock-in method. Their $^1$H- and $^{13}$C-NMR spectra were in good agreement with the reported data[35,36].

### ABA measurement in plants

Extraction, purification, and quantification of ABA were carried out as described in the previous report[71] using around 5-mg dry weights of around 10-day-seedling roots per replicate (equivalent to around 260 (Mock or Ct4)-320 (Ct3) roots).

### Mineral measurements

Plant samples were dried for 3 days in a 70 °C oven. Dried samples were set as 10-15 mg in one biological sample. Each sample was digested with nitric acid and hydrogen peroxide. The dried pellets, after digestion, were dissolved in 0.08 M $HNO_3$. Elemental concentrations in samples were measured by inductively coupled plasma mass spectrometry (ICP-MS) Agilent 7800 (Agilent Technologies Co., Ltd., Japan) according to the manufacturer's instructions. To determine the concentration of P, ICP-MS NexION 350 S (PerkinElmer, Waltham, MA, USA) was used. Clustering analysis was done by using mineral concentrations of the samples. NbClust package (version 3.0) of R was used for k-means clustering (Hartigan-Wong algorithm). An optimal number of clusters was determined by silhouette score.

## Sucrose measurement in plants

Sucrose extraction from the freeze-dried sample and its derivatization for GC/MS analysis were essentially carried out as previously described[71]. GC-EIMS used in this study was a 7890B GC-MS system (Agilent Technologies) equipped with a DB-5 MS + Dura Guard (30 m × 0.25 mm i.d., film thickness of 0.5 μm and 10 m Dura Guard, Agilent Technologies). The injection temperature was 250 °C, and the helium gas flow rate through the column was 0.9 mL/min. The column temperature was held at 60 °C for 1 min and then was raised by 10 °C/min to 325 °C and was held there for 10 min isothermally. The retention time for sucrose-8TMS was 16.4 min under the condition.

## Reporting summary

Further information on research design is available in the Nature Portfolio Reporting Summary linked to this article.

## Data availability

Accession codes: Newly obtained fungal whole-genome data have been deposited in DDBJ/NCBI (Ct61: BQXX01, Ct3: BQXV01, Ct4: BQXW01, KHC23: BPPY01, *C. liriopes*: BPPX01). Additionally, raw sequencing data of RNA-seq transcriptomic data have been deposited in DDBJ (DRA012868, DRA012867). Microscopic pictures have been deposited in BioImage Archived (S-BIAD827). Source data are provided in this paper.

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

## Acknowledgements

We thank Mie Matsubara, Akemi Uchiyama, Shunsuke Imai, Takafumi Shimizu, Sachiko Minezaki, and Hiromi Haba for their technical assistance. We thank Paul Schulze-Lefert, Soledad Sacristán, Ryo Tabata, Chika Tateda, Momoko Takagi, Yuri Tajima, and Ren Ujimatsu for the fruitful discussion. We thank Yasuyuki Kubo, Maarten Koornneef, Javier Paz-Ares, Shigetaka Yasuda, Haruhiko Inoue, and Richard O'Connell for published materials. This work was supported in part by the JSPS KAKENHI Grant (16H06279, 18K14466, 18H04822, 19H05688, 20H02986, 21H05150, and 22H02204 (A.M.)), the JST grant (JPMJPR16Q7, JPMJCR19S2, JPMJSC1702, and JPMJFR200A) and The Uehara Memorial Foundation (A.M.). Computations were partially performed on the NIG supercomputer at ROIS National Institute of Genetics.

## Author contributions

K.H. initiated the project. K.H. and Y.S. directed the research. K.H., J.T., T.H., A.S., M.O., N.K., M.N., N.K., Y.O., R.S., and K.T. conducted the experiments. Y.U. conducted comparative genomic analyses except for molecular phylogenetic tree analyses. S.A. conducted analyses of molecular phylogenetic trees, supervised by W.I. K.H. conducted RNA-seq analyses. J.T. conducted a metabolites analysis supervised by H.O. and A.M. T.S. provided unique material. K.H., S.A., T.H., M.O., Y.U., N.K., M.N., Y.O., R.S., K.T., A.M., and W.I. analyzed the data. K.H., S.A., W.I., and Y.S. wrote the paper with feedback from all authors.

## Competing interests

The authors declare no competing interests.
