## [Peer Review File · Nature Communications]

A fungal sesquiterpene biosynthesis gene cluster critical for mutualist-pathogen transition in *Colletotrichum tofieldiae*Reviewer #1 (Remarks to the Author):

The manuscript by Hiruma and colleagues investigates the relevance of notably abscisic acid and plant nutrition in plant-fungus interactions. They focus on different isolates of the fungus *Colletotrichum tofieldiae* (Ct), which have different effects on plant fitness. Some isolates promote growth while another is pathogenic. The authors identify the underlying genetic traits that confer this variation in interaction types as gene clusters encoding ABA and another secondary metabolite botrydial.

The study is exciting by shedding light on the genetic mechanisms whereby plant-associated fungi can interact with their plant host as either mutualists or pathogens. The authors have included in planta and in-vitro experiments as well as genome and transcriptome analyses to unravel the molecular interactions of Ct isolates with *Arabidopsis thaliana* genotypes, including mutants.

While I find the topic of great relevance, I have some concerns with the presented study:

First of all, the main conclusion in the manuscript is not well supported. Based on RNAseq data and fungal mutants the authors conclude that expression of the gene clusters ABA-BOT is necessary for fungal virulence and that this is, in part, mediated via manipulation of host nutrition. ABA also regulates growth in fungi. Thereby the fungal aba deletion mutants generated here, may be impaired in other processes unrelated to virulence and plant interaction. The correct way of demonstrating the relevance of these fungal secondary metabolite gene clusters would be by heterologous expression.

I miss a biological interpretation of the conclusions related to Ct3-mediated virulence. Wouldn't a pathogen prevent its own growth and reproduction if it shut down nutrient uptake of the host? The authors must take their conclusion further and discuss what the fitness advantage could be in this case.

It is not clear how the authors confirm infection of the different Ct strains in roots. The manuscript mentions that fluorescence microscopy is done (L. 558), however, I could not find trace of those microscopy analyses in the manuscript.

The authors must demonstrate that inoculation of the different Ct strains confer root colonization, and ideally also demonstrate which root compartments are colonized by pathogenic and growth promoting isolates. They obtain fungal transcripts from root extracts, however the fungal hyphae may as well be on the surface of plant roots.

The Ct61 isolate is more than 11% diverged from the two other Ct isolates. This sounds like extensive divergence and raises the question if this is another species. The extent of rearrangements shown in Figure 1e is also dramatic and indicates that these isolates do not belong to the same species. The authors should provide further evidence that Ct61 is indeed the same species as Ct3 and Ct4.

In this regard, Figure 1F does not make sense. The tree indicates that Ct3, Ct4 and Ct61 are equally distant to one another and in fact very closely related. Either the 11% of genome divergence is restricted to particular regions of the genome or the resolution in the tree based on Beta tubulin, GAPDH and ACT is misleading. The authors must provide convincing information about the phylogenetic and genomic relationship of the three isolates.

Microscopy analyses are crucial also for the RNAseq experiment. How do the authors know that roots are infected or not, as they compare gene expression levels? In this regard, the authors also need to explain how samples were collected for RNA extraction. Gene expression patterns, including ABA content, may differ greatly across different root compartments. Therefore, ensuring consistency in the sampling is necessary and the authors must explain how they obtained comparable root tissue from the different Ct treated and mock plants.

It would be excellent to see the presumably neutral KHC strain in the different assays presented in figure 1. There is so far not data showing that this strain is indeed neutral. The authors should present comparable results demonstrating the similarity of mock to KHC inoculation. This could also include genome data.

The manuscript presents an impressive amount of results. It would be good to see these summarized in a more schematic cartoon/figure to illustrate the proposed pathways and functions that are effected by beneficial and pathogenic fungi.

L. 75-78: I am not sure about the statement made here, that all infectious fungi show high intra-species variation. The authors refer to a paper that only studies one fungus, namely *Aspergillus fumigatus*. The authors should either remove this statement or provide sufficient references to justify it.

Figure 1A: The image of liquid cell cultures is of poor resolution. The authors should provide better images eventually quantitative data to demonstrate the different growth morphologies.

Figure 1C and 1D: It is unclear why the three Ct strains are not consistently included in the analyses. In this regard, the authors should include Ct61 in Figure 1C and Ct4 and Ct61 in Figure 1D to highlight the particular effects caused by infection with Ct3. The data is somewhat included in the supplementary material and it would be good to see the phenotypes presented next to each other.

L. 553: Can the author guarantee that there has been no heterologous insertion of the deletion construct? A Southern blot could be applied to confirm correct integration of solely one construct.

L. 610: Why have the authors aligned ABA and BOT genes (and I assume the authors mean proteins here?) against the *Homo sapiens* genome assembly?

Suppl. Fig 3: I believe, it would be more informative if the authors show the protein alignment of ABA and BOT genes.

Reviewer #2 (Remarks to the Author):

The manuscript submitted by the authors and entitled: "A fungal secondary metabolism gene cluster enables mutualist-pathogen transition in root endophyte *Colletotrichum tofieldiae*" reveal a *C. tofieldiae* strain (Ct3) that severely inhibits plant growth. The authors claim that Ct3 pathogenesis occurs through activation of host abscisic acid (ABA) pathways via a fungal secondary metabolism gene cluster related to sesquiterpene ABA and botrydial (BOT) biosynthesis. The manuscript is confusing, and a part of the problem is given by the title that is catching but misleading

1. Is the "activation" of a secondary metabolism gene cluster that cause a different phenotype and not the cluster itself
2. No transition has been shown... a transition means that an organism change lifestyle during time (e.g. Ct3 first promote plant growth and then become pathogenic); here the authors show that different strains have different lifestyles

Another issue that makes the manuscript really confusing is the use of acronyms such as ABA and BOT (used for molecules, clusters, genes, mutant, plant, fungus, etc..) an example of how things can be made much easier is adding the initials of the organisms to a gene and keep the accepted nomenclature where genes are in small letters and italics (eg *Bc-aba1* or *Ct-aba1*, etc...)

Not considering the confusion there are some aspects that could have been better exploited (all linked to each other's) related to the lack of knowledge in fungal metabolisms:

1. This work is focused on a fungal secondary metabolite gene cluster; however no evidence is presented about different metabolic profiles in Ct3 vs Ct4 for example and Ct3 vs the a deletion mutant is presented. Especially because the SMGs and SMGCs are complex and disruption of one single gene may reflect in lots of changes in the metabolic profiles
2. Nothing is mentioned about the structure of the clusters, for example is well known that fungal SMGCs have backbone and tailoring genes... what are the genes in this cluster or in those clusters?
3. No complementation of the KO has been performed

However my main concern is: there has been a huge rearrangement between ABA and BOT

clusters in *B. cinerea* and *Colletotrichum* suggesting the rise of a new metabolic pathway and products. Why are the authors assuming that those are still two different clusters related to ABA and BOT? DO they have biological evidence of the production of those two metabolites?

Reviewer #3 (Remarks to the Author):

It is important for bio-controls in agricultural production to fully understand the regulatory mechanisms of pathogenesis-mutualism transition of plant-associated fungi. This work by Hiruma et al has provided tremendous genetic evidences, trying to convince mainly that: 1. activation and biosynthesis of sesquiterpene ABA and botrydial (BOT) in *Colletotrichum tofieldiae* (Ct) is one of the pathways for its pathogenesis on *Arabidopsis thaliana*, otherwise this fungus should be mutualistic; 2. The pathogenesis of this fungus on the plant also occurs through activation of the host ABA biosynthesis; 3. The pathogenesis from the fungus affects the host nutrition states. Overall, this work has shown some new insights of the pathogenic mechanisms of Ct on At, from the point of view of fungal/plant secondary metabolism, while this fungus traditionally is mutualistic. However, most conclusions are drawn from the genetic data, which need more work to confirm, and some descriptions have been overstated. In addition, the overall logic is not good as for the data presented.

Some major concerns:

1. Authors have emphasized the mutualist-pathogen transition of fungi in this manuscript (such as in the title), and shown brief introduction of this transition (first paragraph of Introduction). However, authors did not show the real mutualist-pathogen transition of Ct in this work (without altering the fungal genomic sequence) or other examples of fungi, except the genetically modified Ct3 strain (*aba*, *bot* deletions). So I cannot accept this logic that ABA/BOT are required for the transition, and it only means that ABA/BOT are possibly required for the pathogenesis of Ct.
2. In Introduction, there is no introduction for transition mechanism, or ABA involvement in signaling. No any reports? They are important for this work.
3. Based on transcriptomic screening, this manuscript suggested that overexpressed ABA gene cluster or overproduction of ABA in Ct3 is responsible for its pathogenesis, along with bulky genetic data. However, I have some suggestions: 1) ABA genes are only among the 304 early-inducible genes, what and how about other possible master genes?; 2) if ABA/BOT are the determinants for pathogenesis of Ct, what happens if ABA is added to Ct4 or other strains. 3) if ABA/BOT gene cluster is genetically put under the control of some growth-phase dependent promoters, Can Ct3 or Ct4 transit between mutualistic-pathogenic states?
4. The possibility has been raised that ABA should be one of the main virulence factors both for the fungus and for the host. It should be explained that how ABA from the fungus affects the host, by ABA-signaling or others? Since depletion of host ABA has abolished the fungal pathogenesis, what is the role of fungal ABA? And what is the difference of ABA from fungus/host?
5. Authors also have made a lot of efforts to reveal the responsive pathways of the host to the Ct3 strain. However, I do not think this work is so necessary to understand the fungal mutualist-pathogen transition. But more work in fungal signaling or fungus-host interaction would be more highly related to understand the transition mechanisms. But most of above are absent.
6. Most conclusions are drawn based on the genetic data, which could only suggest the gene functions/the correlation between genes and phenotypes. I suggest more evidences such as from biochemistry, chemistry, cell biology, to further understand/confirm the mechanisms. Moreover, some descriptions are overstated, such as line 247 (fungal ABA biosynthesis genes mediate host ABA responses.), here we could only genetically conclude that fungal ABA biosynthesis is required for host ABA responses. Here only genetic data were provided, and we could only see the relationship between fungal ABA biosynthesis/host ABA responses. And somewhere also exist such descriptions.
7. Logically speaking, the paragraph (line 125-141) for genome alignment of three Ct strains could not provide significant insights for their various interactions with hosts. I could not see any helpful evidences or conclusions to understand this difference. In my point, it is also the case for the phylogenetic analysis of gene clusters.
8. Too many un-related references, and some of them are not uniform

Response to Reviewer 1

Reviewer #1 (Remarks to the Author):

The manuscript by Hiruma and colleagues investigates the relevance of notably abscisic acid and plant nutrition in plant-fungus interactions. They focus on different isolates of the fungus *Colletotrichum tofieldiae* (Ct), which have different effects on plant fitness. Some isolates promote growth while another is pathogenic. The authors identify the underlying genetic traits that confer this variation in interaction types as gene clusters encoding ABA and another secondary metabolite botrydial.

The study is exciting by shedding light on the genetic mechanisms whereby plant-associated fungi can interact with their plant host as either mutualists or pathogens. The authors have included in planta and in vitro experiments as well as genome and transcriptome analyses to unravel the molecular interactions of Ct isolates with *Arabidopsis thaliana* genotypes, including mutants.

Our response: Thank you very much for your positive evaluation of our work.

While I find the topic of great relevance,

I have some concerns with the presented study: First of all, the main conclusion in the manuscript is not well supported. Based on RNAseq data and fungal mutants the authors conclude that expression of the gene clusters ABA-BOT is necessary for fungal virulence and that this is, in part, mediated via manipulation of host nutrition. ABA also regulates growth in fungi. Thereby the fungal *aba* deletion mutants generated here, may be impaired in other processes unrelated to virulence and plant interaction. The correct way of demonstrating the relevance of these fungal secondary metabolite gene clusters would be by heterologous expression.

Our response: Thanks very much for the advice. We have further examined whether *Ct3 Δaba* and *Ct3 Δbot* mutants exhibit any variations in growth under normal and low Pi media conditions used for our plant inoculation assays. Determination of fungal colony diameters indicates that their culture growth is not affected compared to the wild-type under both conditions, in agreement with the results in nutrient-rich Mathur's media (Supplementary Fig. 7a-c). It was previously shown that *Bc Δaba* or *Bc Δbot* deletion mutants also display wild-type-like growth under standard nutrient-rich media (Siewers et al., 2004; Moraga et al., 2016). The results indicate that the gene clusters ABA-BOT do not have a significant impact on fungal basal growth except during plant colonization. We have incorporated the new data

in the revised **Fig. 6a**. We will implement heterologous expression of the gene clusters in future studies to ensure their functionality.

Siewers V, Smedsgaard J, Tudzynski P. The p450 monooxygenase BcABA1 is essential for abscisic acid biosynthesis in *Botrytis cinerea*. *Appl Environ Microb* 70, 3868-3876 (2004).

Moraga J, et al. Genetic and Molecular Basis of Botrydial Biosynthesis: Connecting Cytochrome P450-Encoding Genes to Biosynthetic Intermediates. *ACS Chem Biol* 11, 2838-2846 (2016).

I miss a biological interpretation of the conclusions related to Ct3-mediated virulence. Wouldn't a pathogen prevent its own growth and reproduction if it shut down nutrient uptake of the host? The authors must take their conclusion further and discuss what the fitness advantage could be in this case.

Our response: From our results together with previous studies, we infer that the perturbation of the nutrient acquisition and utilization of the host via the host ABA pathway, facilitates fungal acquisition of sugars from the host. This is consistent with previous reports that exogenous ABA application leads to increased accumulation of sugar and amino acids, which act as osmolytes retaining water inside the plant tissues (Seki et al., 2007; Urano et al., 2009; Taji et al. 2002). We have incorporated the discussion in the revised manuscript (**See Line 442-443, Fig. 9**).

Seki M, Umezawa T, Urano K, Shinozaki K. Regulatory metabolic networks in drought stress responses. *Curr Opin Plant Biol* 10, 296-302 (2007).

Urano K, et al. Characterization of the ABA-regulated global responses to dehydration in *Arabidopsis* by metabolomics. *Plant J* 57, 1065-1078 (2009).

Taji T, et al. Important roles of drought- and cold-inducible genes for galactinol synthase in stress tolerance in *Arabidopsis thaliana*. *Plant J* 29, 417-426 (2002).

It is not clear how the authors confirm infection of the different Ct strains in roots. The manuscript mentions that fluorescence microscopy is done (L. 558), however, I could not find trace of those microscopy analyses in the manuscript.

The authors must demonstrate that inoculation of the different Ct strains confer root colonization, and ideally also demonstrate which root compartments are colonized by pathogenic and growth promoting isolates. They obtain fungal transcripts from root extracts, however the fungal hyphae may as well be on the surface of plant roots.

Our response: We are grateful for this comment. To address this, we have generated new Ct3 and Ct4 strains that constitutively express GFP through Agrobacterium-mediated transformation, which retain wild-type-like infection phenotypes. Live imaging analyses for the transgenic fungal lines under confocal microscopy demonstrated that both pathogenic and growth-promoting fungi colonize the epidermal and cortical cell layers of *A. thaliana* roots (See revised Fig.1e-f .and Line 129-137, Supplementary Videos 1-2). In our view, the data of fungal transcripts, whether derived from fungal hyphae on the surface or within the roots, give a useful proxy for assessing fungal activity when combined with such imaging data. Future studies will be required to unambiguously determine whether and if so how fungal hyphae in different root compartments contribute to the overall inoculation effects on the host.

The Ct61 isolate is more than 11% diverged from the two other Ct isolates. This sounds like extensive divergence and raises the question if this is another species. The extent of rearrangements shown in Figure 1e is also dramatic and indicates that these isolates do not belong to the same species. The authors should provide further evidence that Ct61 is indeed the same species as Ct3 and Ct4.

In this regard, Figure 1F does not make sense. The tree indicates that Ct3, Ct4 and Ct61 are equally distant to one another and in fact very closely related. Either the 11% of genome divergence is restricted to particular regions of the genome or the resolution in the tree based on Beta tubulin, GAPDH and ACT is misleading. The authors must provide convincing information about the phylogenetic and genomic relationship of the three isolates.

Our response: Since the introduction of 'Genealogical Concordance Phylogenetic Species Recognition' by Taylor et al. in 2000, fungal taxonomy and identification have primarily relied on Multi-Locus Sequence Analysis (MLSA). Currently, genomic data are not utilized in fungal identification. We adopted a relevant methodology to identify *Colletotrichum* species within the spaethianum clade, to which Ct belongs (Vieira et al., 2020). To ensure that these strains indeed belong to the same species, we have additively constructed phylogenetic trees based on 6 fungal genes, following a widely accepted method (Damm et al. 2009; Jayawardena et al. 2021). The results also indicate that these strains are indeed the same species (now in Supplementary Fig. 1). Additional phylogenetic trees based on amino acids of 1509 single-copy genes also indicate that all Ct strains belong to the same monophyletic clade, despite some degree of variations (Fig. 2b). Thus, we infer from the results that these strains belong to the same species.

Regarding the genomic nucleotide sequence divergence among Ct strains, the presentation of median percentages in the figure might have led the reviewer to the comments. We used the Mauve program to compare Ct3, Ct4, and Ct61 identities by pair-aligning their draft genomes. The resulting alignments, termed LCB (locally collinear blocks), have different lengths due to variations in the draft genome quality of each strain. Some LCBs have high identities but are short, while others have low identities but are long. To avoid the potential imprecision by using the mean values of total sequence of LCBs, we decided to use the median identities of the LCBs to obtain one identity value for each comparison.

In this respect, please note that comparative genomic analyses among *F. oxysporum* species also have pointed out large genomic divergences within the same species (e.g., Ma et al., Nature 2012).

Taylor, J.W., Jacobson, D.J., Kroken, S., Kasuga, T., Geiser, D.M., Hibbett, D.S. and Fisher, M.C. (2000). Phylogenetic species recognition and species concepts in fungi. *Fungal Genet. Biol.* 31: 21–32.

Vieira, W., Bezerra, P.A., Silva, A.C.D., Veloso, J.S., Camara, M.P.S., and Doyle, V.P. (2020). Optimal markers for the identification of *Colletotrichum* species. *Mol Phylogenet Evol* 143, 106694.

Damm U, Woudenberg JHC, Cannon PF, Crous PW. *Colletotrichum* species with curved conidia from herbaceous hosts. *Fungal Divers* 39, 45-87 (2009).

Jayawardena RS, Bhunjun CS, Hyde KD, Gentekaki E, Itthayakorn P. *Colletotrichum*: lifestyles, biology, morpho-species, species complexes and accepted species. *Mycosphere* 12, 519-669 (2021).

Ma, L.J., van der Does, H., Borkovich, K. et al. Comparative genomics reveals mobile pathogenicity chromosomes in *Fusarium*. *Nature* **464**, 367–373 (2010). <https://doi.org/10.1038/nature08850>.

Microscopy analyses are crucial also for the RNAseq experiment. How do the authors know that roots are infected or not, as they compare gene expression levels? In this regard, the authors also need to explain how samples were collected for RNA extraction. Gene expression patterns, including ABA content, may differ greatly across different root compartments. Therefore, ensuring consistency in the sampling is necessary and the authors must explain how they obtained comparable root tissue from the different Ct treated and mock plants.

Our response: In this study, we determined the time points based on plant phenotypic variations observed after inoculation with each microbe. As stated above, live imaging

analyses revealed that both pathogenic and beneficial fungi extensively colonize the roots, with largely uniformed root colonization modes for each fungal strain, in our settings. Having made sure these points, we have collected the whole root compartments for the molecular analyses. We have also described the details in the revised methods. (See line 613-614, Supplementary Fig. 9)

It would be excellent to see the presumably neutral KHC strain in the different assays presented in figure 1. There is so far not data showing that this strain is indeed neutral. The authors should present comparable results demonstrating the similarity of mock to KHC inoculation. This could also include genome data.

Our response: We have shown the data that KHC did not inhibit plant growth under low Pi in the revised Supplementary Fig. 3b. We have also rephrased the term “neutral” to “nonpathogenic”, which we think is relevant in the context.

Although we initially considered KHC strain within the Ct species, additional phylogenetic analyses with the six marker genes and the entire genomes revealed that this strain is not closely related to Ct, as shown in the revised Supplementary Fig. 1 and Fig. 2b. Since we are currently conducting a separate project on this KHC strain and its relatives isolated from naturally-grown Brassicaceae plants in Japan, we wish to reserve further information regarding KHC for the future work.

The manuscript presents an impressive amount of results. It would be good to see these summarized in a more schematic cartoon/figure to illustrate the proposed pathways and functions that are effected by beneficial and pathogenic fungi.

Our response: We have added a schematic figure to summarize our study in the revised Fig. 9.

L. 75-78: I am not sure about the statement made here, that all infectious fungi show high intra-species variation. The authors refer to a paper that only studies one fungus, namely *Aspergillus fumigatus*. The authors should either remove this statement or provide sufficient references to justify it.

Our response: We have removed this statement.

Figure 1A: The image of liquid cell cultures is of poor resolution. The authors should provide better images eventually quantitative data to demonstrate the different growth morphologies.

Our response: We have presented high-resolution images of fungal colonies and the quantitative data in the revised **Fig. 1a and Supplementary Fig. 2a**. The original images were not obtained from liquid cell cultures, but from solid cultures in flasks. In the revision, we used solid cultures on plates.

Figure 1C and 1D: It is unclear why the three Ct strains are not consistently included in the analyses. In this regard, the authors should include Ct61 in Figure 1C and Ct4 and Ct61 in Figure 1D to highlight the particular effects caused by infection with Ct3. The data is somewhat included in the supplementary material and it would be good to see the phenotypes presented next to each other.

Our response: Due to the constraints of our plant infection assay apparatus, we focus on Ct3 and Ct4 as the primary fungal strains under examination in this study. Following the reviewer's recommendation, we have incorporated the Ct4 data in the revised **Fig. 1d**.

L. 553: Can the author guarantee that there has been no heterologous insertion of the deletion construct? A Southern blot could be applied to confirm correct integration of solely one construct.

Our response: We were not concerned with possible heterologous insertions of the deletion construct, since we have evaluated multiple (2-4) independent lines for each knockout mutant. Moreover, instead of Southern blot analyses, we have shown that *Ct3BOT5* complementation lines reverted the mutant phenotypes to the wild-type in the revised **Supplementary Fig. 7I and the text (Line 340-342)**. We thus conclude that the observed phenotypes result from the disruption of these genes.

L. 610: Why have the authors aligned ABA and BOT genes (and I assume the authors mean proteins here?) against the Homo sapiens genome assembly?

Our response: Yes, we aligned their protein sequences with those from *Homo sapiens*. This is because P450s are well annotated in the *Homo sapiens* genome assembly. This figure is moved to the revised Fig.8a and Supplementary Fig. 8a.

Suppl. Fig 3: I believe, it would be more informative if the authors show the protein alignment of ABA and BOT genes.

Our response: We agree with the reviewer's suggestion. We present the protein sequence alignment of CtABA and CtBOT. (**Supplementary Fig. 4o**)

Response to Reviewer 2

Reviewer #2 (Remarks to the Author):

The manuscript submitted by the authors and entitled: "A fungal secondary metabolism gene cluster enables mutualist-pathogen transition in root endophyte *Colletotrichum tofieldiae*" reveal a *C. tofieldiae* strain (Ct3) that severely inhibits plant growth. The authors claim that Ct3 pathogenesis occurs through activation of host abscisic acid (ABA) pathways via a fungal secondary metabolism gene cluster related to sesquiterpene ABA and botrydial (BOT) biosynthesis.

The manuscript is confusing, and a part of the problem is given by the title that is catching but misleading

1. Is the "activation" of a secondary metabolism gene cluster that cause a different phenotype and not the cluster itself

Our response: Thank you for this helpful comment. We have revised the title as follows.

'A fungal abscisic acid and botrydial gene cluster critical for mutualist-pathogen transition in *Colletotrichum tofieldiae*'

2. No transition has been shown... a transition means that an organism change lifestyle during time (e.g. Ct3 first promote plant growth and then become pathogenic); here the authors show that different strains have different lifestyles

Our response: The initial manuscript presented the data that Ct3 indeed transitions between pathogenic and beneficial lifestyles in a temperature-dependent manner. Your comments however suggest that we should stress this point in the revision. At 22 °C, Ct3 exhibits pathogenesis through ABA and BOT genes, but at 26 °C when ABA-BOT expression is lowered, Ct3 promotes plant growth under low Pi conditions. Notably, the suppression of Ct pathogenesis at 26 °C is dependent on *A. thaliana* AtPHR1/AtPHL1, central regulators of plant phosphate starvation responses. Although the underlying mechanisms warrants further investigation, the data clearly demonstrate that ABA-BOT contributes to fungal lifestyle transition in Ct3. Furthermore, we show that Ct3-mediated plant growth inhibition was drastically reduced at 24 dpi compared with at 10 dpi even at 22 °C, especially under low Pi (Supplementary Fig. 2g), which is also associated with ABA-BOT cluster silenced at 24 dpi (Supplementary Table 7). These imply that this strain transits the different lifestyles during time. This also provides evidence that Ct3 transits the lifestyles during the infection time. We

have relocated the temperature-related data to the revised **Fig. 7 and the text (Line 345-361)** from the original Figure 3 and Figure S3.

Another issue that makes the manuscript really confusing is the use of acronyms such as ABA and BOT (used for molecules, clusters, genes, mutant, plant, fungus, etc..) an example of how things can be made much easier is adding the initials of the organisms to a gene and keep the accepted nomenclature where genes are in small letters and italics (eg *Bc-aba1* or *Ct-aba1*, etc...)

Our response: Thanks very much for your kind advice. We have modified the fungal gene names as suggested through the text, e.g. *Ct3ABA1* gene or *Ct3Δaba1* mutant. We have also included *At* before the gene name of *A. thaliana* genes (e.g., *AtPHR1*, *AtAT4*).

Not considering the confusion there are some aspects that could have been better exploited (all linked to each other's) related to the lack of knowledge in fungal metabolisms:

1. This work is focused on a fungal secondary metabolite gene cluster; however no evidence is presented about different metabolic profiles in Ct3 vs Ct4 for example and Ct3 vs the a deletion mutant is presented. Especially because the SMGs and SMGCs are complex and disruption of one single gene may reflect in lots of changes in the metabolic profiles

Our response: In this respect, we conducted some experiments to detect metabolites derived from Ct3. The detail is described below.

2. Nothing is mentioned about the structure of the clusters, for example is well known that fungal SMGCs have backbone and tailoring genes... what are the genes in this cluster or in those clusters?

Our response: Thank you for your suggestions. We have added some information for the ABA/BOT biosynthetic genes/enzymes. Revised **Supplementary Table 6a** summarizes their functions of the corresponding gene products. Proposed biosynthetic pathways of ABA/BOT clusters are shown in the revised **Fig. 5 and Supplementary Fig. 6**. In addition, amino acid sequences were compared between the identified *CtABA* (*ABA1-4*) and *CtBOT* (*BOT1-7*) genes in Ct3 and their corresponding *BcABA* and *BcBOT* genes in *B. cinerea*. This indicates that each of the *CtABA* and *CtBOT* genes is closely related to the corresponding *BcABA* and *BcBOT* genes, as shown in the revised **Supplementary Table 6b**.

Product	Gene (B. cinerea)	Gene (C. tofieldiae)	putative function	biosynthetic role
abscisic acid	bcABA1	ctABA1	cytochrome P450	tailoring enzyme
	bcABA2	ctABA2	cytochrome P450	tailoring enzyme
	bcABA3	ctABA3	alpha-ionylideneethane synthase	sesquiterpene core formation
	bcABA4	ctABA4	short-chain dehydrogenase/reductase	tailoring enzyme
botrydial	bcBOT1	ctBOT1	cytochrome P450	tailoring enzyme
	bcBOT2	ctBOT2	presilphiperfolan-8-beta-ol synthase	sesquiterpene core formation
	bcBOT3	ctBOT3	cytochrome P450	tailoring enzyme
	bcBOT4	ctBOT4	cytochrome P450	tailoring enzyme
	bcBOT5	ctBOT5	acetyltransferase	tailoring enzyme
	bcBOT6	ctBOT6	transcription factor	-
	bcBOT7	ctBOT7	dehydrogenase	tailoring enzyme

Supplementary Table 6a

(A) Proposed ABA biosynthetic pathway

(B) Proposed BOT biosynthetic pathway

Scheme. Proposed biosynthetic pathways of (A) abscisic acid (ABA) and (B) botrydial (BOT).

We have incorporated these pathways in **Fig. 5 and Supplementary Fig. 6.**

3. No complementation of the KO has been performed

Our response: We have shown complementation of the *Ct3*Δ *bot5* mutant with *Ct3BOT5* gene, which restores WT-like phenotypes in the *Ct3* mutant in the revised **Supplementary Fig. 7I with text (Line 340-342)**. We chose this mutant because it showed the greatest inoculation effects on the plant host among the tested *Ct3* mutant lines under our tested conditions.

However my main concern is: there has been a huge rearrangement between ABA and BOT clusters in *B. cinerea* and *Colltotrichum* suggesting the rise of a new metabolic pathway and products. Why are the authors assuming that those are still two different clusters related to ABA and BOT? DO they have biological evidence of the production of those two metabolites?

Our response:

Thank you for pointing out this important issue. In the original manuscript, we hypothesized that *Ct3ABA* and *Ct3BOT* genes act in separate metabolic pathways for ABA and BOT biosynthesis, respectively, assuming on the distinct phenotypes of the *Ct3*Δ *aba3* and *Ct3*Δ *bot5* mutants as shown in **Fig. 6e**.

To test the hypothesis, we profiled metabolites from *Ct3*, *Ct3* mutants, and *Ct4*, in collaboration with Drs Atsushi Minami, Junya Takino, and Hideaki Oikawa (Hokkaido University), who have studied on the ABA and BOT biosynthetic pathways using heterologous expression system in *Aspergillus oryzae* (e.g., Takino et al., *J. Am. Chem. Soc.* 2018). By focusing on the BOT biosynthetic intermediates, 4β-acetoxypobotryan-9β-ol and 4β-acetoxypobotryane-9β,15α-diol, we observed those compounds in *Ct3* strain only when incubated in rice-based media. Importantly, the production of these metabolites is completely abolished in the deletion mutant of an acetyltransferase gene *Ct3BOT5*, whereas *Ct3*Δ *aba2* and *Ct3*Δ *aba3* mutant strains still produced those metabolites. These results suggested that BOT biosynthetic genes in *Ct3* strain are specifically involved in the biosynthesis of BOT. In contrast, no ABA or related compounds were not detected under any of the tested conditions, suggesting that ABA biosynthesis in *Ct3* only occurs under specific conditions, as also inferred from the detection of 4β-acetoxypobotryan-9β-ol and 4β-acetoxypobotryane-9β,15α-diol being limited to the rice-based media. These additional experimental results are summarized in the revised **Fig. 5, Supplementary Figs. 5-6 with text (Line 290-306)**.

In silico analysis also supported our hypothesis. All the ABA/BOT biosynthetic enzymes in Ct3 exhibit > 50% amino acid sequence identities with the corresponding enzymes in *B. cinerea* despite their distant phylogenetic relationship. This is consistent with the results that Ct3 strain produced BOT biosynthetic intermediates. Of particular note, no gene encoding a typical tailoring enzyme, such as a cytochrome P450, a flavoprotein oxidase, and a short-chain dehydrogenase/reductase, is annotated within the 20,000 bp region. Taken together, we believe that new modifications leading to ABA/BOT derivatives are unlikely.

Takino J, et al. Unveiling Biosynthesis of the Phytohormone Abscisic Acid in Fungi: Unprecedented Mechanism of Core Scaffold Formation Catalyzed by an Unusual Sesquiterpene Synthase. *J Am Chem Soc* 140, 12392-12395 (2018).

Response to Reviewer 3

Reviewer #3 (Remarks to the Author):

It is important for bio-controls in agricultural production to fully understand the regulatory mechanisms of pathogenesis-mutualism transition of plant-associated fungi. This work by Hiruma et al has provided tremendous genetic evidences, trying to convince mainly that: 1. activation and biosynthesis of sesquiterpene ABA and botrydial (BOT) in *Colletotrichum tofieldiae* (Ct) is one of the pathways for its pathogenesis on *Arabidopsis thaliana*, otherwise this fungus should be mutualistic; 2. The pathogenesis of this fungus on the plant also occurs through activation of the host ABA biosynthesis; 3. The pathogenesis from the fungus affects the host nutrition states. Overall, this work has shown some new insights of the pathogenic mechanisms of Ct on At, from the point of view of fungal/plant secondary metabolism, while this fungus traditionally is mutualistic. However, most conclusions are drawn from the genetic data, which need more work to confirm, and some descriptions have been overstated. In addition, the overall logic is not good as for the data presented.

Our response: Thank you for your positive evaluation. In this revised manuscript, we have incorporated additional data to corroborate the genetic data and reorganized the figures and text to improve the clarity. Regarding “the overstatement” of the fungal lifestyle transition, we believe that the revisions strengthen the conclusions.

Some major concerns:

1. Authors have emphasized the mutualist-pathogen transition of fungi in this manuscript (such as in the title), and shown brief introduction of this transition (first paragraph of Introduction). However, authors did not show the real mutualist-pathogen transition of Ct in this work (without altering the fungal genomic sequence) or other examples of fungi, except the genetically modified Ct3 strain (aba, bot deletions). So I cannot accept this logic that ABA/BOT are required for the transition, and it only means that ABA/BOT are possibly required for the pathogenesis of Ct.

Our response: As replied to the Reviewer 2, we have clarified this point in the revised manuscript. Please see above and the **revised Fig. 7 and related text (Line 345-361)**.

2. In Introduction, there is no introduction for transition mechanism, or ABA involvement in signaling. No any reports? They are important for this work.

Our response: We have added some information about the possible involvement of fungal-derived ABA in the modulation of host ABA signaling and responses. (See Line 73-80)

3. Based on transcriptomic screening, this manuscript suggested that overexpressed ABA gene cluster or overproduction of ABA in Ct3 is responsible for its pathogenesis, along with bulky genetic data. However, I have some suggestions: 1) ABA genes are only among the 304 early-inducible genes, what and how about other possible master genes?;

Our response: The reviewer is correct. There are other genes that seem interesting. Among them, we put an initial focus on ABA/BOT in the present study. As it will probably require some years of work to examine an additional target gene(s), we wish to explore these genes in future studies.

2) if ABA/BOT are the determinants for pathogenesis of Ct, what happens if ABA is added to Ct4 or other strains.

Our response: We have shown that exogenous ABA application inhibited *A. thaliana* shoot growth and Ct4-mediated PGP under low Pi, suggesting that ABA negatively affects plant growth under low Pi, in **Supplementary Fig. 7j with text (Line 336-339)**.

3) if ABA/BOT gene cluster is genetically put under the control of some growth-phase dependent promoters, Can Ct3 or Ct4 transit between mutualistic-pathogenic states?

Our response: Thank you for this interesting suggestion. As replied to the Reviewer 2 above, we describe that *Ct3ABA* and *Ct3BOT* gene expression depends on the temperature and host *AtPHR1/AtPHL1* function in the revised manuscript (**Fig. 7**). We would like to undertake the proposed approach using several candidate promoters in the near future.

4. The possibility has been raised that ABA should be one of the main virulence factors both for the fungus and for the host. It should be explained that how ABA from the fungus affects the host, by ABA-signaling or others? Since depletion of host ABA has abolished the fungal pathogenesis, what is the role of fungal ABA? And what is the difference of ABA from fungus/host?

Our response: Based on our genetic, metabolite and plant gene expression profiles, we posit that fungal ABA and related metabolites, such as botrydial, activate host ABA core pathways, resulting in inhibition of several nutrient pathways and severe growth inhibition of

the host plant (fungal pathogenesis). We have added some statement in the revised discussion to address these points **(Fig. 9, See line 482-493)**.

One possible explanation for the reduced pathogenesis observed in ABA-depleted *A. thaliana* mutants (such as *aba2* mutant plants) is that host-derived ABA also contributes to the activation of responses induced by fungal-derived ABA and related metabolites. We have added texts and a model to address these points in the revised discussion **(See line 482-493, Fig. 9)**. It was previously shown that the structure of fungal-derived ABA is identical to that of plants, despite distinct biosynthetic pathways, in different fungi (e.g., Assante et al., 1977; Marumo et al., 1982; Takino et al., 2019; Drama et al., 2019). This information has been added in the revised introduction **(See line 73-75)**.

Assante G, Merlini, L, Nasini,G. (+)-Abscisic acid, a metabolite of the fungus *Cercospora rosicola*. *Experientia* 33, 1556-1557 (1977).

Marumo S, Katayama M, Komori E, Ozaki Y, Natsume M, Kondo S. Microbial-Production of Abscisic-Acid by *Botrytis-Cinerea*. *Agr Biol Chem Tokyo* 46, 1967-1968 (1982).

Takino J, et al. Unveiling Biosynthesis of the Phytohormone Abscisic Acid in Fungi: Unprecedented Mechanism of Core Scaffold Formation Catalyzed by an Unusual Sesquiterpene Synthase. *J Am Chem Soc* 140, 12392-12395 (2018).

Darma R, Lutz A, Elliott CE, Idnurm A. Identification of a gene cluster for the synthesis of the plant hormone abscisic acid in the plant pathogen *Leptosphaeria maculans*. *Fungal Genet Biol* 130, 62-71 (2019).

5. Authors also have made a lot of efforts to reveal the responsive pathways of the host to the Ct3 strain. However, I do not think this work is so necessary to understand the fungal mutualist-pathogen transition. But more work in fungal signaling or fungus-host interaction would be more highly related to understand the transition mechanisms. But most of above are absent.

Our response: Thank you for this suggestion. We think that this work is significant, as it reveals host responses to Ct3 (or host manipulation by Ct3) as a key step in fungal pathogenesis expression, central to mutualist-pathogen transition. We agree with the reviewer for the importance of elucidating the fungal mechanisms involved, but we wish to address them in the future.

6. Most conclusions are drawn based on the genetic data, which could only suggest the gene functions/the correlation between genes and phenotypes. I suggest more evidences such as from biochemistry, chemistry, cell biology, to further understand/confirm the mechanisms.

Our response: In response to the reviewer's comments, we have conducted the following additional experiments.

(1) Measurements and Analyses of Ct3-derived metabolites

As replied to Reviewer 2 above, we conducted metabolite analysis in Ct3, Ct3 mutants, and Ct4 in collaboration with Drs Atsushi Minami, Junya Takino, and Hideaki Oikawa (Hokkaido University), who have studied on the ABA and BOT biosynthetic pathways using heterologous expression system in *Aspergillus oryzae* (e.g., Takino et al., *J. Am. Chem. Soc.* 2018). The results are seen in the revised in **Fig. 5, Supplementary Figs. 5-6 with text (Line 290-306)**.

Takino J, et al. Unveiling Biosynthesis of the Phytohormone Abscisic Acid in Fungi: Unprecedented Mechanism of Core Scaffold Formation Catalyzed by an Unusual Sesquiterpene Synthase. *J Am Chem Soc* 140, 12392-12395 (2018).

(2) Live imaging analyses for Ct3 and Ct4 during the root colonization.

As replied to Reviewer 1 above, we have traced the root colonization of GFP-expressing Ct3 and Ct4 strains under microscopy, showing that both pathogenic and beneficial Ct strains colonize at least the epidermal and cortical cell layers of *A. thaliana* roots. Together with the ABA and BOT biosynthetic gene expression during the initial root colonization, these microscopic analyses suggest that Ct3 expresses ABA-BOT in the intra- and inter-cellular hyphae in roots.

Moreover, some descriptions are overstated, such as line 247 (fungal ABA biosynthesis genes mediate host ABA responses.), here we could only genetically conclude that fungal ABA biosynthesis is required for host ABA responses. Here only genetic data were provided, and we could only see the relationship between fungal ABA biosynthesis/host ABA responses. And somewhere also exist such descriptions.

Our response: We have modified these descriptions as the reviewer suggests.

7. Logically speaking, the paragraph (line 125-141) for genome alignment of three Ct strains could not provide significant insights for their various interactions with hosts. I could not see any helpful evidences or conclusions to understand this difference.

Our response: We agree with the reviewer's comment. We have reduced the description of genome alignment of three Ct strains, but kept the basic genome information in the revised manuscript. In our view, this is important because the detailed characterization of these Ct strains (especially Ct3 and Ct4) has been conducted for the first time in this study.

In my point, it is also the case for the phylogenetic analysis of gene clusters.

Our response: As Reviewer 1 points out, it is important to discuss the evolution of ABA-BOT gene clusters and phylogenetic relationships between the Ct strains examined. As replied to Reviewer 1, we have strengthened the phylogenetic analysis of each strain, as seen in the revised **Fig. 1 and Supplementary Fig. 1**. Although the results may not provide direct insight into the mechanisms underlying their differences, we hope the information gives an important basis for future studies.

8. Too many un-related references, and some of them are not uniform

Our response: We have removed the references not directly related to this work.

Reviewer #1 (Remarks to the Author):

The authors have replied to the comments raised by myself and the other reviewers. Hereby also new data and analyses have been added to the manuscript. I notably appreciate the metabolite profiling which provide evidence for a role of the botrydial sesquiterpene BOT. It was not possible to detect the fungal biosynthesis of ABA during different tested conditions. Could it be that the gene cluster encodes for another metabolite?

One of my previous points referred to colonization of Arabidopsis roots. The authors have conducted microscopy analyses and provide images of an mCherry-tagged membrane associated aquaporin. The new data presented in supplementary movies 1 and 2, does not convince me that the inter-cellular space is colonized by the fungus. The highlighted spots on Figure 1 can be many things. I am puzzled that there is no hypha growing along root cells? If the indicated spots are entry points of hyphae, why do we not observe further growth in the roots?

The evidence for horizontal gene transfer is very sparse. In fact, just comparing the structure of the gene clusters is in-sufficient to conclude that the sequences have been horizontally transferred. If the authors do not decide to conduct correct inference of a possible horizontal gene transfer event, I propose to exclude this statement.

In my previous review, I was concerned about the absence of southern blot analyses to confirm the correct locus-transformation and the absence of ectopic integration. The authors explain in the response letter that they have conducted a complementation test instead. Validating the correct transformation is not the same as generating a complementation test as mutants with ectopic integration may exhibit other phenotypes, under other conditions. So, my concern has not been addressed. Moreover, in Supplementary Figure 7I, I do not see complementation tests for all mutants. The complementation strains are also not named in the correct way to show that they indeed are complementation strains. I may be missing some details here?

L. 99-100: change text to "... five different Ct strains...."

L. 103: Change sentence here: "... these five strains exhibited different growth morphology during in vitro growth...."

L. 117: "persistence" is not the right term here. Please rephrase, e.g. use the phrasing "...suggests a pathogenic lifestyle in associated with a broad diversity of Brassicaceae species"

L. 125-126: I do not agree that the authors can "infer" the relevance of sugars. It is more correct to say "... we hypothesize that...."

L. 135: It is more appropriate here to say "... colonize the inter-cellular space"
You can also use apoplastic space.

L. 150-151: This sentence would benefit from some re-phrasing. It is not clear what is meant by "as proxies..."? As proxies for what?

L337: The authors do not show that the predicted ABA cluster, can confer the biosynthesis of ABA in vitro as well as in planta. Thus I would reformulate here to say that "... the fungal derived ABA may play a role..."

L 345: I believe the right term is "shift between lifestyles" in stead of "transits"

Figure 1: I think the supplementary figure 2C is more convincing in terms of showing the effect of Ct3. I suggest using this type of image showing the whole-plant from the side to introduce the distinct effects of Ct3 on plant growth.

Figure 2C: What is KHC on the tree?

Figure 8A: It would be good to see the tree with an outgroup. This would confirm the close relatedness of the Ct and Botrytis genes.

Figure S4m: Please also indicate which genes are used to produce the species tree.

Reviewer #2 (Remarks to the Author):

Following the reviewer's comments the authors did improve a lot the manuscript that is now much less confused and more linear.

The authors have also filled the gap between genetics and the secondary metabolites produced. that was my main concerns and the manuscript have been improved also in this aspect.

One minor thing is related to Figure 2: "Genomic analyses of beneficial and pathogenic Ct strains" I could not find the list of genomes / data used in Figure 2b; please make sure all the accession numbers and the references of the genomes used are reported.

Reviewer #3 (Remarks to the Author):

The manuscript in the current status has been improved and partially answered my concerns. But two major conceptual questions still remain.

(1) How can authors make a convincing conclusion that Ct3 and other Ct strain are in the same species? As shown below, based on the phylogenetic data and comparative genome analysis, I can not fully accept that they belong to the same species. This conclusion is the foundation for all studies and discussion in this paper.

Line 101, how can authors make a conclusion that they belong to the same species? It is extremely important for the paper to describe the transition mechanism. I suggest to have more discussion and draw a careful conclusion on it.

(2) The main novelty in this paper is the finding of ABA/BOT for control of transition between mutualism-pathogenesis of Ct strains. So authors should put more emphasis on the mechanism for this transition (the regulatory mechanisms of Ct3 for its transition, but not abundant work on host responses, though it is necessary). Though transition data have been presented after temperature shift for Ct3, but no more novel molecular insights (except bot gene expression). So I would say again as in the first review, author can have more work on the mechanism study of fungal transition. Though ABA/BOT have been shown involved in the pathogenesis, as other reviewers raised, these genetic data can not fully convey this concept.

Other concerns for data presentation in some examples:

Fig. 1b. it is not clear about the significant difference represented by different letters.

Fig.1d. as for the Mock group, there is a big variation when testing Ct3 and Ct4. With Pi, about 600 VS 500; without Pi, about 800 VS 1900. Is there any system error for this variation? How can authors make it convincing for other testing data?

Response to Reviewer#1

Reviewer #1 (Remarks to the Author):The authors have replied to the comments raised by myself and the other reviewers. Hereby also new data and analyses have been added to the manuscript. I notably appreciate the metabolite profiling which provide evidence for a role of the botrydial sesquiterpene BOT. It was not possible to detect the fungal biosynthesis of ABA during different tested conditions. Could it be that the gene cluster encodes for another metabolite?

Our response: Thank you very much for your positive comments. Given the significant conservation observed in the ABA biosynthetic genes between distantly related *Colletotrichum* and *Botrytis*, it is reasonable to infer that the ABA gene cluster mediates ABA production. This hypothesis is further supported by the data that the application of exogenous ABA successfully complemented the *Ct3Aaba3* mutant phenotypes. However, it cannot be ruled out that the gene cluster is involved in production of another metabolite(s) than ABA. We have acknowledged these possibilities in the legend of Figure 9.

One of my previous points referred to colonization of Arabidopsis roots. The authors have conducted microscopy analyses and provide images of an mCherry-tagged membrane associated aquaporin. The new data presented in supplementary movies 1 and 2, does not convince me that the inter-cellular space is colonized by the fungus. The highlighted spots on Figure 1 can be many things. I am puzzled that there is no hypha growing along root cells? If the indicated spots are entry points of hyphae, why do we not observe further growth in the roots?

Our response: We agree with you that the presence of hyphae growing within intercellular spaces is perplexing. Movies 1 and 2, along with Figures 1e and 1f, demonstrate an invasion of host cells by hyphae in the establishment of biotrophic phases. In this revision, we have included additional microscopic images illustrating the elongation of Ct3 and Ct4 hyphae within the intercellular and intracellular spaces of host cells (Figure 1g and h).

The evidence for horizontal gene transfer is very sparse. In fact, just comparing the structure of the gene clusters is in-sufficient to conclude that the sequences have been

horizontally transferred. If the authors do not decide to conduct correct inference of a possible horizontal gene transfer event, I propose to exclude this statement.

Our response: Thank you very much for your comment. We realize the need for additional clarification regarding the horizontal gene transfer of BOT biosynthetic clusters. In this revision, we have revised the following paragraph. Please note that the phylogenetic trees presented in Figure 4 and Supplementary Fig. 4 comprise fungal species that are distantly related, therefore, each branch of the tree can be interpreted as horizontal rather than vertical transfer. We have thus discussed the matter as follows.

Line 270-279:

The substantial differences between species and gene trees suggest horizontal gene transfers (HGTs) distributing the ABA and BOT biosynthesis gene clusters among these plant-associated fungi. Our comparative genomic analysis revealed that *Diaporthe helianthi*, a plant pathogen, also possesses a single ABA-BOT biosynthesis gene cluster of high synteny when compared with Ct (Supplementary Fig. 4n). **On an assumption that the ABA-BOT cluster originated only once according to the principle of parsimony, the results suggest a HGT of the BOT cluster to *Botrytis*,** (Fig. 4d and Supplementary Fig. 4m–n). Given that the ABA and BOT clusters have likely arisen multiple times in different phytopathogenic fungi, it is conceivable that the acquisition of ABA and BOT biosynthesis gene cluster(s) contributes to pathogenesis evolution in plant-associated fungi.

In my previous review, I was concerned about the absence of southern blot analyses to confirm the correct locus-transformation and the absence of ectopic integration. The authors explain in the response letter that they have conducted a complementation test instead. Validating the correct transformation is not the same as generating a complementation test as mutants with ectopic integration may exhibit other phenotypes, under other conditions. So, my concern has not been addressed. Moreover, in Supplementary Figure 7l, I do not see complementation tests for all mutants. The complementation strains are also not named in the correct way to show that they indeed are complementation strains. I may be missing some details here?

Our response: Thank you for your comments. We acknowledge that we have not provided sufficient explanation in this regard. In the previous study, we introduced the wild-type BOT5 genes, along with their corresponding regulatory sequence, into the *Ct3Abot5* knockout mutant strains (we have modified the name of the lines as +BOT5# in *Ct3Abot5*). Consequently, we observed a recovery of the mutant phenotype towards the wild type, as depicted in Figure 7I. We acknowledge the reviewer's observation that we did not conduct this procedure for all the mutants due to time constraints. However, we would like to highlight that we have generated knockout fungal mutants for at least two distinct ABA or BOT biosynthetic genes located in different positions within the cluster (e.g., *Ct3Aaba2* and *Ct3Aaba3*), and we have consistently observed reduced pathogenicity across all mutants. In light of these findings, we believe that performing the complementation assay on the *Ct3Abot5* mutants, which exhibit the clearest phenotypes under our experimental conditions, is sufficient for the purpose of this study.

L. 99-100: change text to "... five different Ct strains....L. 103: Change sentence here: "... these five strains exhibited different growth morphology during in vitro growth....

Our response: Modified as suggested.

L. 117: "persistence" is not the right term here. Please rephrase, e.g. use the phrasing "...suggests a pathogenic lifestyle in associated with a broad diversity of Brassicaceae species"

Our response: Modified as suggested.

L. 125-126: I do not agree that the authors can "infer" the relevance of sugars. It is more correct to say "... we hypothesize that...."

Our response: Modified as suggested.

L. 135: It is more appropriate here to say "... colonize the inter-cellular space" You can also use apoplastic space.

Our response: We have added additional images showing that Ct forms both intra- and inter-cellular hyphae (Fig.1g-h).

L. 150-151: This sentence would benefit from some re-phrasing. It is not clear what is meant by "as proxies..."? As proxies for what?

Our response: we have modified as follows. However, a molecular phylogenetic analysis using the conserved 1509 single-copy genes among the tested 64 fungal species suggests that beneficial Ct strains have evolved from its pathogenic relatives, such as *C. incanum* (Fig. 2b).

L337: The authors do not show that the predicted ABA cluster, can confer the biosynthesis of ABA in vitro as well as in planta. Thus I would reformulate here to say that "... the fungal derived ABA may play a role..."

Our response: Modified as suggested.

L 345: I believe the right term is "shift between lifestyles" in stead of "transits"

Our response: Modified as suggested.

Figure 1: I think the supplementary figure 2C is more convincing in terms of showing the effect of Ct3. I suggest using this type of image showing the whole-plant from the side to introduce the distinct effects of Ct3 on plant growth.

Our response: we have moved the supplementary Figure 2C to the main Figure. 1. Instead, the previous Figure 1C is moved to Supplementary Fig. 2C.

Figure 2C: What is KHC on the tree?

Our response: we have described KHC as *C. higginsianum* KHC in the tree.

Figure 8A: It would be good to see the tree with an outgroup. This would confirm the close relatedness of the Ct and Botrytis genes.

Our response: We have added an outgroup in the tree. We have replaced the previous tree without the outgroup to the new tree with the outgroup. We have added additional information in Figure 8a as follow.

Maximum Likelihood (ML) tree of ABA and BOT genes in Cytochrome P450 family using IQ-TREE version 1.6.11. Ct3: *C. tofieldiae* Ct3; Bc: *B. cinerea* B05.10; Hs: *Homo sapiens*. Homologs to CYP7A1 were used as an outgroup. The phylogenetic relationship of the ABA and BOT genes and other 409 P450 genes in *C. tofieldiae*, *B. cinerea*, and *H. sapiens* is indicated in Fig. S4o. Ultrabootstrap probability is shown on the branches. The scale bar represents substitutions per site.

Figure S4m: Please also indicate which genes are used to produce the species tree.

Our response: We have followed Li et al. (2021) for the species tree. We have added the reference in the list. In addition, we have added some explanations in the method as follow.

The fungal species tree in Fig.4m is followed by⁶⁴.

Reference

Li, Y., Steenwyk, J.L., Chang, Y., Wang, Y., James, T.Y., Stajich, J.E., Spatafora, J.W., Groenewald, M., Dunn, C.W., Hittinger, C.T., Shen, X-X., and Rokas, A. (2021) A genome-scale phylogeny of the kingdom Fungi. *Curr Biol* 31, 1653–1665.

Response to Reviewer#2

Reviewer #2 (Remarks to the Author): Following the reviewer's comments the authors did improve a lot the manuscript that is now much less confused and more linear. The authors have also filled the gap between genetics and the secondary metabolites produced. that was my main concerns and the manuscript have been improved also in this aspect. One minor thing is related to Figure 2: "Genomic analyses of beneficial and pathogenic Ct strains" I could not find the list of genomes / data used in Figure 2b; please make sure all the accession numbers and the references of the genomes used are reported.

Our response: We have provided the information in the revised Supplementary Table 3.

Response to Reviewer#3

Reviewer #3 (Remarks to the Author): The manuscript in the current status has been improved and partially answered my concerns. But two major conceptual questions still remain. (1) How can authors make a convincing conclusion that Ct3 and other Ct strain are in the same species? As shown below, based on the phylogenetic data and comparative genome analysis, I cannot fully accept that they belong to the same species. This conclusion is the foundation for all studies and discussion in this paper. Line 101, how can authors make a conclusion that they belong to the same species? It is extremely important for the paper to describe the transition mechanism. I suggest to have more discussion and draw a careful conclusion on it.

Our response: Thank you for raising this significant point. We have previously addressed the concern regarding our conclusion that these Ct strains belong to the same species in our earlier reply to the reviewer. Having acknowledged the need for further explanation, we have provided additional explanation in the revised Line 101-103 and Line 440-446.

Our previous response with additional information (the lines highlighted are added here) :

‘ **Our response:** Since the introduction of 'Genealogical Concordance Phylogenetic Species Recognition' by Taylor et al. in 2000, fungal taxonomy and identification have primarily relied on Multi-Locus Sequence Analysis (MLSA). Currently, genomic data are not utilized in fungal identification (we have also checked the recently published papers for this). We adopted a relevant methodology to identify *Colletotrichum* species within the spaethianum clade, to which Ct belongs (Vieira et al., 2020). To ensure that these strains indeed belong to the same species, we have additively constructed phylogenetic trees based on 6 fungal genes, following a widely accepted method (e.g., Damm et al. 2009; Damm et al., 2012; Sato et al., 2015; Hacquard et al., 2016; Damm et al., 2019; Damm et al., 2020; Jayawardena et al. 2021; Liu et al., 2022). The results also indicate that these strains are indeed the same species (**Supplementary Fig. 1**). Additional phylogenetic trees based on amino acids of 1509 single-copy genes also indicate that all Ct strains belong to the same monophyletic clade, despite some degree of variations (**Fig. 2b**). Thus, we infer from the results that these strains belong to the same species.

Regarding the genomic nucleotide sequence divergence among Ct strains, the presentation of median percentages in the figure might have led the reviewer to the comments. We used the Mauve program to compare Ct3, Ct4, and Ct61 identities by pair-aligning their draft genomes. The resulting alignments, termed LCB (locally collinear blocks), have different lengths due to variations in the draft genome quality of each strain. Some LCBs have high identities but are short, while others have low identities but are long. To avoid the potential imprecision by using the mean values of total sequence of LCBs, we have originally decided to use the median identities of the LCBs to obtain one identity value for each comparison.

In this respect, please note that comparative genomic analyses among *F. oxysporum* species also have pointed out large genomic divergences within the same species (e.g., Ma et al., Nature 2012).

References

Taylor, J.W., Jacobson, D.J., Kroken, S., Kasuga, T., Geiser, D.M., Hibbett, D.S. and Fisher, M.C. (2000). Phylogenetic species recognition and species concepts in fungi. *Fungal Genet. Biol.* 31: 21–32.

Vieira, W., Bezerra, P.A., Silva, A.C.D., Veloso, J.S., Camara, M.P.S., and Doyle, V.P. (2020). Optimal markers for the identification of *Colletotrichum* species. *Mol Phylogenet Evol* 143, 106694.

Damm U, Woudenberg JHC, Cannon PF, Crous PW. *Colletotrichum* species with curved conidia from herbaceous hosts. *Fungal Divers* 39, 45-87 (2009).

.

Damm, U., Cannon, P.F., Woudenberg, J.H.C. and Crous, P.W. (2012). The *Colletotrichum acutatum* species complex. *Stud Mycol* 73, 37-113.

Damm, U., O'Connell, R.J., Groenewald, J.Z., and Crous, P.W. (2014). The *Colletotrichum destructivum* species complex – hemibiotrophic pathogens of forage and field crops. *Stud Mycol* 79, 49–84.

Sato T, Moriwaki J, Kaneko S. Anthracnose Fungi with Curved Conidia, *Colletotrichum* spp. belonging to Ribosomal Groups 9-13, and Their Host Ranges in Japan. *Jarq-Jpn Agr Res Q* 49, 351-362 (2015).

Hacquard, S., Kracher, B., Hiruma, K., Munch, P.C., Garrido-Oter, R., Thon, M.R., Weimann, A.,

Damm, U., Dallery, J.F., Hainaut, M., et al. (2016). Survival trade-offs in plant roots during colonization by closely related beneficial and pathogenic fungi. *Nat Commun* 7, 11362.

Damm U, Sato T, Alizadeh A, Groenewald, J.Z., and Crous, P.W. (2019). The *Colletotrichum dracaenophilum*, *C. magnum* and *C. orchidearum* species complexes. *Stud Mycol* 92, 1–46.

Damm U, Sun YC, Huang CJ (2020). *Colletotrichum eriobotryae* sp. nov. and *C. nymphaeae*, the anthracnose pathogens of loquat fruit in central Taiwan, and their sensitivity to azoxystrobin. *Mycological Progress* 19, 367–380.

Jayawardena RS, Bhunjun CS, Hyde KD, Gentekaki E, Itthayakorn P. *Colletotrichum*: lifestyles, biology, morpho-species, species complexes and accepted species. *Mycosphere* 12, 519-669 (2021).

Liu, F., Ma, Z.Y., Hou, L.W., Diao, Y.Z., Wu, W.P., Damm, U., Song, S., and Cai, L. (2022). Updating species diversity of *Colletotrichum*, with a phylogenomic overview. *Stud Mycol* 101, 1–56.

Ma, L.J., van der Does, H., Borkovich, K. *et al.* Comparative genomics reveals mobile pathogenicity chromosomes in *Fusarium*. *Nature* **464**, 367–373 (2010). <https://doi.org/10.1038/nature08850>.

(2) The main novelty in this paper is the finding of ABA/BOT for control of transition between mutualism-pathogenesis of Ct strains. So, authors should put more emphasis on the mechanism for this transition (the regulatory mechanisms of Ct3 for its transition, but not abundant work on host responses, though it is necessary). Though transition data have been presented after temperature shift for Ct3, but no more novel molecular insights (except bot gene expression). So, I would say again as in the first review, author can have more work on the mechanism study of fungal transition. Though ABA/BOT have been shown involved in the pathogenesis, as other reviewers raised, these genetic data cannot fully convey this concept.

Our response: Thank you for bringing this issue to our attention. While we acknowledge its importance, we recognize that providing a comprehensive answer would entail a substantial amount of additional work that requires years of time.

Nonetheless, we concur with the reviewer's suggestions to include more fungal data to facilitate the discussion of potential regulatory mechanisms involving ABA-BOT in Ct3.

1. To address this issue, we have conducted supplementary analyses and integrated pertinent statements into the discussion section, recognizing the necessity for subsequent inquiries in future investigations. We have explained our logical flow for identifying ABA and BOT genes. These genes exhibited co-regulation alongside other factors (total 92 genes including ABA and BOT genes) during pathogenesis. Their expression was exclusive to the pathogenic phase (at 10 dpi), while being entirely repressed at a subsequent time point when the Ct3-mediated inhibition of plant growth was notably diminished (at 24 dpi). Please also check Line 232-235.
2. We have conducted additional analyses on the fungal-side RNAseq data (as summarized in Supplementary Table 12) to investigate the potential influence of ABA-BOT on these putative (virulence) factors described in point 1. Our findings indicate that the expression of these 92 Ct3 genes was not largely affected by defects in ABA or BOT biosynthetic genes (Supplementary Table 13), implying that ABA-BOT is not involved in regulating other genes and that the expression status of ABA-BOT plays a crucial role in the manifestation of virulence. Please also check Line 369-376.
3. We have included additional statements to highlight the necessity of future investigations in unraveling the regulatory mechanisms underlying the modulation of Ct3 ABA-BOT, as shown below. Please also check Line 440-446.

Notably, the investigated Ct strains, albeit classified into the Ct species through the widely accepted method for *Colletotrichum* species, harbor substantial genomic sequence variations. Although there are not significant sequence differences in the ABA-BOT region, there are subtle variations outside the ABA-

BOT region between beneficial and pathogenic Ct strains, which might contribute to the divergence in the expression profiles of ABA-BOT genes and in fungal infection modes. Further studies are warranted to examine this hypothesis and the underlying mechanisms.

Other concerns for data presentation in some examples:

Fig. 1b. it is not clear about the significant difference represented by different letters.

Our response: We have confirmed that these different letters represent significant difference. Having met the reviewer's comments, we have decided to modify the figure to clearly show that all the Ct strains except Ct3 increase plant biomass under low Pi.

Fig.1d. as for the Mock group, there is a big variation when testing Ct3 and Ct4. With Pi, about 600 VS 500; without Pi, about 800 VS 1900. Is there any system error for this variation? How can authors make it convincing for other testing data?

Our response: We have found that our calculation was wrong. Thanks very much for pointing out this error. We have now modified the Figure correctly.

Reviewer #3 (Remarks to the Author):

This version of revised manuscript has been significantly improved, and I suggest to publish.